# Improved MPC for trajectory planning of self-driving cars

Kai Sun[1], Niaona Zhang[1*], Zonghao Li[2], Haitao Ding[2], Heshu Zhang[1], Changhong Jiang[1*]

**1** Electrical and Electronic Engineering, Changchun University of Technology, Changchun, Jilin Province, China, **2** College of Automotive Engineering, Jilin University, Changchun, Jilin Province, China

☯ These author contributed equally to this work.
* zhangniaona@163.com (NZ); jch@ccut.edu.cn (CJ)

**Data availability statement:** The data underlying the results presented in the study

## Abstract

To tackle the challenges associated with lane changes in diverse driving environments, this study introduces an Improved Model Predictive Control (IMPC) approach that integrates vehicle state and driving context to execute maneuvers effectively. Initially, a simplified model is established, incorporating vehicle position, velocity, and driving conditions. Subsequently, a Sigmoid function is utilized to restrict vehicle movements, while Finite State Machine (FSM) decision-making selects appropriate maneuvers in real time according to changing driving conditions. In addition, a discrete Simplified Dual Neural Network (SDNN) is utilized to efficiently solve Quadratic Programming (QP) problems, overcoming the slow-solving issue of traditional MPC and ensuring rapid acquisition of longitudinal and lateral accelerations for lane changes, braking, and overtaking maneuvers. Finally, the results from hardware-in-the-loop (HIL) simulation validation indicate that IMPC can efficiently address collision avoidance with surrounding dynamic vehicles across three distinct driving scenarios.

## Introduction

In recent years, advancements in vehicle technology and artificial intelligence have fostered the emergence of autonomous vehicles. Data indicate that many road traffic accidents stem from driver negligence and delayed reactions. Self-driving vehicles could significantly enhance road safety, enhance passenger comfort, and decrease driver workload [1,2]. Furthermore, they can effectively mitigate these incidents. Consequently, research on autonomous vehicles has become pivotal, emphasizing collision avoidance and overtaking maneuvers under varying driving conditions. This paper introduces the IPMC method to tackle these challenges.

When performing obstacle avoidance and overtaking tasks, autonomous vehicles can determine their path by utilizing road conditions and surrounding vehicle information collected by sensors. Additionally, they adapt the planned path in response to changes in the surrounding environment [3]. Typically, completing obstacle avoidance and overtaking maneuvers comprises two main components: decision-making and control. The decision-making component generates appropriate maneuver decisions, which are subsequently transmitted to the control component. The control component executes these maneuvers while adhering to the constraints of vehicle dynamics and environmental conditions,

are available from"github: https://github.com/s147896325/k.git"and"figshare: 10.6084/m9.figshare.28021790".

**Funding:** This research was funded by NZ of the National Natural Science Joint Fund Project, grant number 52477041; NZ of the Open Fund of the State Key Laboratory of Automotive Simulation and Control of Jilin University, grant number 20210237; and NZ of the Jilin Province Science and Technology Development Plan Project, grant number 20230508049RC. The funders had no role in study design, data collection and analysis, decision to publish, or preparation of the manuscript.

**Competing interests:** There are no financial conflicts of interest to disclose.

thereby accomplishing the obstacle avoidance and overtaking tasks [4]. In summary, an obstacle avoidance and overtaking system consists of two interrelated yet independent parts: path planning and trajectory tracking.

The methods of path planning [5,6] comprise graph-based search algorithms (e.g., Dijkstra's algorithm [7] and A* algorithm [8]), local planning algorithms (e.g., artificial potential field method [9] and dynamic window approach [10]), sampling-based algorithms (e.g., Rapidly-exploring Random Trees [11]), parameter optimization methods (e.g., Bezier curve method [12] and Sigmoid method [13]), and numerical optimization methods. This study utilizes the Improved Model Predictive Control (IMPC) methodology within numerical optimization methods. MPC [14] can integrate road boundary constraints, vehicle dynamics constraints, and drivable area constraints as unified constraints within the prediction process. Moreover, MPC can allocate weights effectively based on the lane-changing context. During the optimization process, more accurate prediction results can be obtained by adjusting the prediction model, objective function, and constraints (e.g., vehicle dynamics and physical limitations). Additionally, future output predictions can be enhanced by adjusting the prediction and control horizons [15–18].

Many scholars have explored the challenges of lane-changing and overtaking in autonomous driving systems. Alrifaee [19] introduced two linear constraints: the Forward Collision Constraint (FCC) and the Rearward Collision Constraint (RCC). These constraints incorporate surrounding vehicles and lane boundaries as barriers within the MPC framework to define feasible driving regions for the ego vehicle. Nilsson [20,21] proposed a method that utilizes a structured environment for formulating low-complexity rolling horizon planning, effectively addressing the high computational complexity associated with trajectory planning. Coskun [22] conducted reachability analysis on surrounding vehicles using dynamic models to predict their future movements accurately. Jalalmaab [23] employed constant safety distances in constraint settings to ensure vehicle safety, although enhancing flexibility in lane-changing under complex scenarios is required. Brown [24] and Esterle [25] utilized enveloping and combinatorial reasoning methods to generate collision-free areas that involve multiple MPC controllers, ensuring stability in individual control strategies, yet necessitating enhanced computational efficiency for joint control scenarios. Combining the strengths and weaknesses of these studies, this paper introduces a function graph akin to a Sigmoid function used for overtaking trajectories. Traditionally employed for overtaking trajectories, the Sigmoid function [26–29] here acts as a constraint barrier to prevent collisions, offering time-varying navigable regions for moving vehicles. A decision variable is assigned to the Sigmoid function. When overtaking is not feasible, braking and deceleration are initiated until the left lane is safely navigable. When encountering obstructed vehicles, scenarios involving both lane-changing and braking are considered, necessitating the introduction of a decision-making function. Various decision-making methods applicable to autonomous driving [30]. Markov decision processes [31] can simulate agents making stochastic decisions in environments where system states exhibit Markov properties. Q-Learning [32] optimizes agent behavior strategies through learning over state-action pairs. Finite state machines (FSM) model a discrete mathematical approach to studying a finite number of states and transitions between them. Due to their simplicity in handling basic behaviors such as lane changing and braking, FSM [33] is chosen as the decision-making algorithm for its ease of implementation, real-time responsiveness, and simplicity of application. In this study, FSM-generated activation signals are combined with the sigmoid function to impose different constraint conditions on Model Predictive Control (MPC).

During vehicle operation, prompt responses are necessary when encountering obstructed vehicles. However, the iterative optimization speed of MPC is insufficient to meet the

real-time and rapid requirements of vehicle movement. The solving process of MPC fundamentally entails online solving of constrained quadratic programming problems. Enhanced MPC integrates a discretized simplified dual neural network that converts constraint intervals into set-point controls through steady-state optimization, thereby effectively reducing computational burdens. Lastly, the neural network resolves the Quadratic Programming (QP) problem, thereby significantly enhancing MPC's solving speed.

The contributions of this paper are summarized as follows:

1. Introducing the Sigmoid function as a constraint barrier to regulate the travel range of the autonomous vehicle. This innovation allows for travel planning without the need to generate specific travel trajectories.

2. Integrating the activation function into MPC constraints involves generating it from the FSM decision, which governs lane changes and braking. This integration reduces the number of constraints and selects optimal maneuvers for specific driving scenarios.

3. Introducing discrete SDNN to solve MPC reduces computation time and enhances the response time of self-driving vehicles.

The structure of this paper is depicted in Fig 1. Section 2 establishes the model. Section 3 addresses the obstacle avoidance trajectory planning problem, (1) encompassing the setup of vehicle driving conditions; (2) behavioral decision-making employing the FSM algorithm; (3) pre-optimization of parameters for the Sigmoid function; (4) establishment of constraints and objective functions; (5) Generation of Sigmoid Constraint Barriers Section 4 presents the discrete SDNN for rapid QP problem-solving. Section 5 conducts simulation analyses under three distinct scenarios. Section 6 concludes the paper and provides future perspectives.

## Model development

Different vehicle models are suitable for various modeling scenarios, and the choice of model significantly influences final prediction outcomes [34]. For example, simpler scenarios require straightforward vehicle models to prevent excessive parameterization that could degrade response times. In this study, MPC is employed to predict the trajectory of the ego vehicle for collision-free overtaking planning. The prediction process focuses exclusively on the longitudinal and lateral velocity, position, and acceleration, treating both the ego vehicle and surrounding vehicles as point masses in motion. Therefore, a point mass model is employed in this study to predict the motion of both the ego vehicle and surrounding vehicles. This model simplifies vehicle dynamics by disregarding size-related information and load transfer due to longitudinal and lateral accelerations, describing vehicle motion as that of a single mass point. Such models are commonly used in the path-planning phase of autonomous vehicles, effectively reducing the computational complexity in trajectory planning [35] (Fig 2).

The kinematic equations for the point mass model are as follows:

$$
\begin{cases}
\ddot{y} = a_y \\
\ddot{x} = a_x \\
\dot{\varphi} = \dfrac{a_y}{a_x} \\
\dot{Y} = \dot{x}\sin\varphi + \dot{y}\cos\varphi \\
\dot{X} = \dot{x}\cos\varphi + \dot{y}\sin\varphi
\end{cases}
\tag{1}
$$

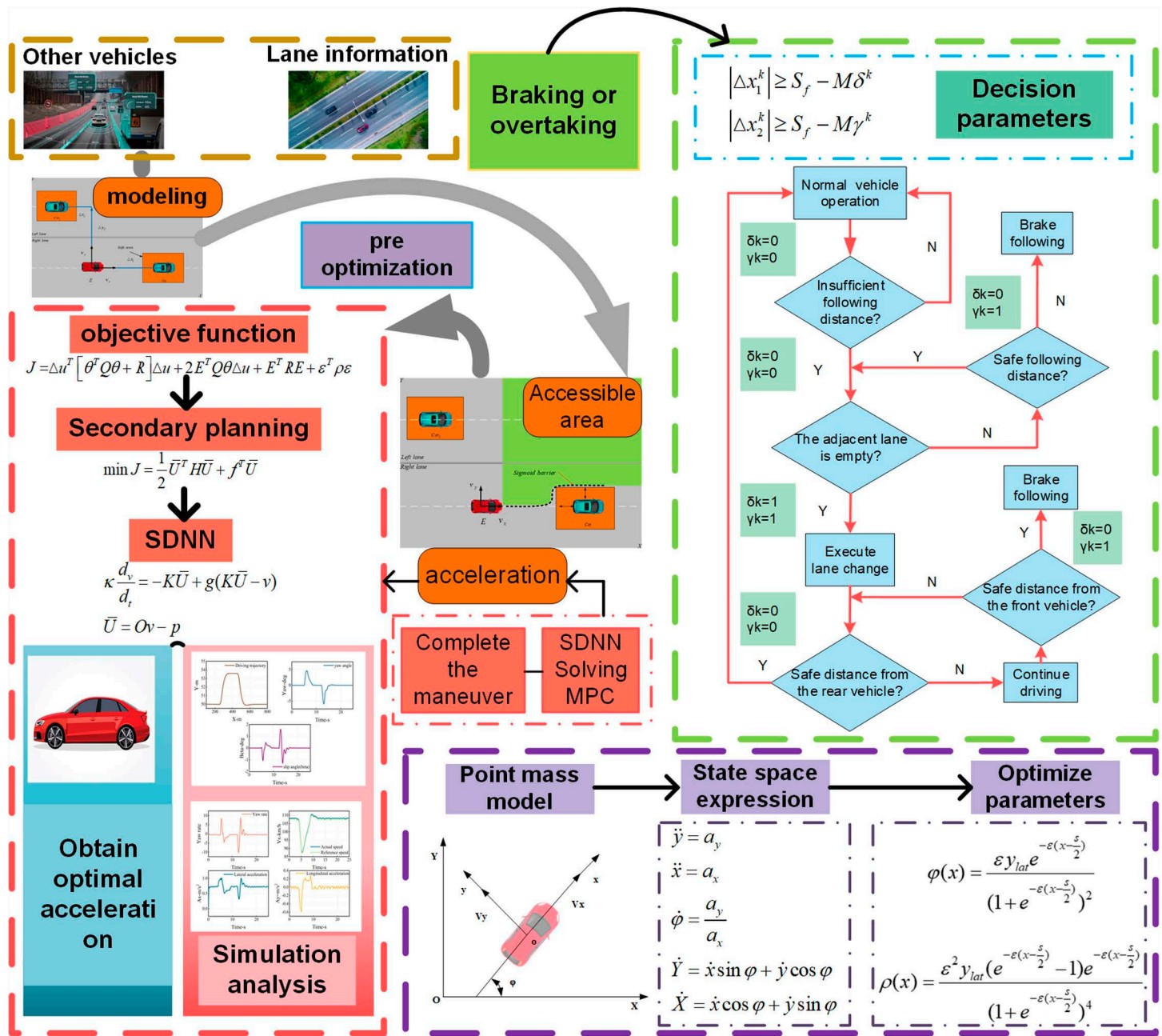

**Fig 1. IMPC Trajectory Planning Framework.** Green, red, and purple respectively denote the decision-making, modeling, and trajectory-planning components.

Where, $x$ and $y$ represent the vehicle's longitudinal and lateral directions in the vehicle body coordinate system, while $X$ and $Y$ denotes the geodetic coordinate system, with $\varphi$ representing the vehicle's heading angle. From the equation above, this system can be considered a control system with input $u = [a_x\ a_y]^T$ and state variable $\chi = [y\ x\ \varphi\ Y\ X]^T$ The general form of this system can be expressed as:

$$\chi = f(\chi, u) \tag{2}$$

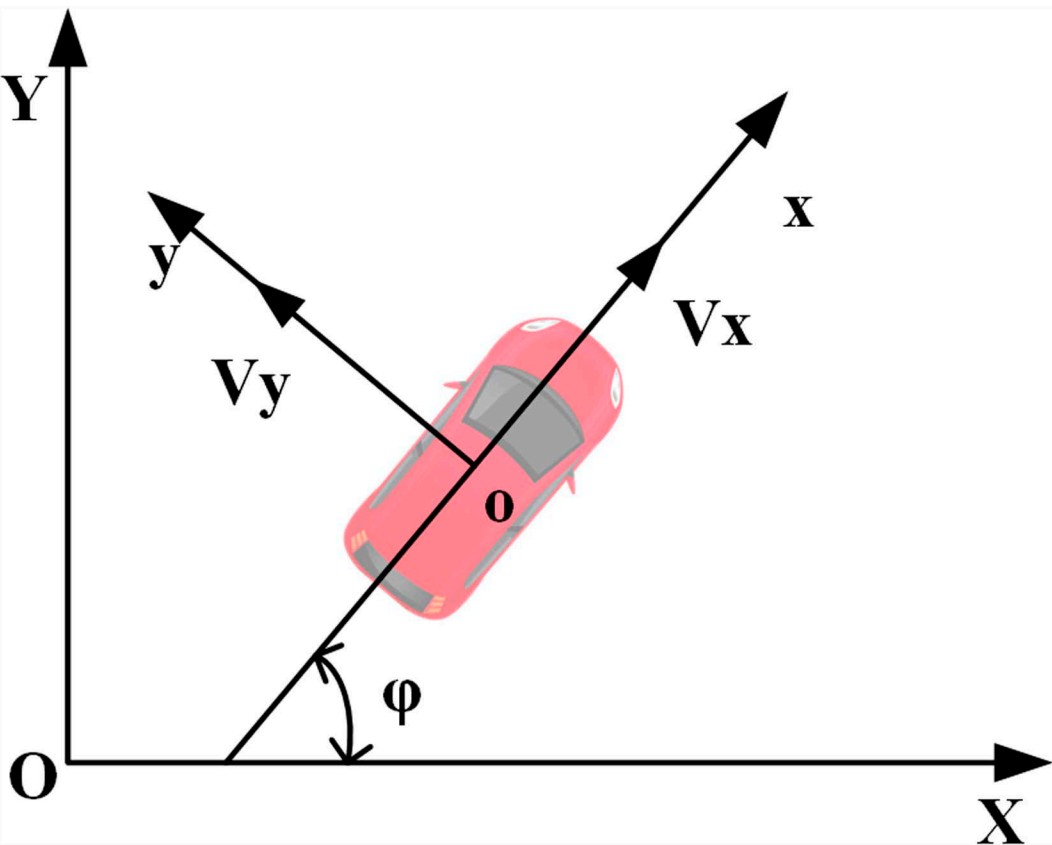

**Fig 2. Automotive point mass model.** $(x, y)$ denotes the vehicle coordinate system, while $(X, Y)$ represents the global coordinate system.

Simply put:

$$\dot{\tilde{\chi}} = A\tilde{\chi} + B\tilde{u} \tag{3}$$

The forward Eulerian discretization yields:

$$\tilde{\chi}(k+1) = \overline{A}\tilde{\chi}(k) + \overline{B}\tilde{u}(k) \tag{4}$$

Where

$$\overline{A} = \begin{bmatrix} 1 & 0 & 0 & 0 & 0 \\ 0 & 1 & 0 & 0 & 0 \\ 0 & 0 & 1 & 0 & 0 \\ \cos\varphi & \sin\varphi & \dot{x}\cos\varphi - \dot{y}\sin\varphi & 1 & 0 \\ -\sin\varphi & \cos\varphi & -\dot{x}\sin\varphi - \dot{y}\cos\varphi & 0 & 1 \end{bmatrix}, \quad \overline{B} = \begin{bmatrix} 1 & 0 \\ 0 & 1 \\ \dfrac{1}{a_x} & -\dfrac{a_y}{a_x^2} \\ 0 & 0 \\ 0 & 0 \end{bmatrix}$$

Transforming the equation results in

$$\xi(k|t) = \begin{bmatrix} \dot{\chi}(k|t) \\ \dot{u}(k-1|t) \end{bmatrix} \tag{5}$$

The resulting state space expression is:

$$\xi(k+1|t) = \tilde{A}\xi(k|t) + \tilde{B}\triangle U(k|t)$$
$$\eta(k|t) = \tilde{C}\xi(k|t) \tag{6}$$

Where $\tilde{A} = \begin{bmatrix} \bar{A} & \bar{B} \\ 0 & I_m \end{bmatrix}$, $\tilde{B} = \begin{bmatrix} \bar{B} \\ I_m \end{bmatrix}$, $n$ represents the control variables, and $m$ represents the state variables.

The predicted output is expressed as:

$$Y(k) = \phi_k \xi(k) + \theta_k \triangle U(k) \tag{7}$$

Where:

$$Y(k) = \begin{bmatrix} \eta(k+1) \\ \eta(k+2) \\ \cdots \\ \eta(k+N_c) \\ \cdots \\ \eta(k+N_p) \end{bmatrix}, \; \phi_k = \begin{bmatrix} \tilde{C}\tilde{A} \\ \tilde{C}\tilde{A}^2 \\ \cdots \\ \tilde{C}\tilde{A}^{N_c} \\ \cdots \\ \tilde{C}\tilde{A}^{N_p} \end{bmatrix}, \; \theta_k = \begin{bmatrix} \tilde{C}\tilde{B} & 0 & 0 & 0 \\ \tilde{C}\tilde{A}\tilde{B} & \tilde{C}\tilde{B} & 0 & 0 \\ \cdots & \cdots & \ddots & \cdots \\ \tilde{C}\tilde{A}^{N_c-1}\tilde{B} & CA^{N_c-2}B & \cdots & CB \\ \tilde{C}\tilde{A}^{N_c}\tilde{B} & \tilde{C}A^{N_c-1}\tilde{B} & \cdots & \tilde{C}\tilde{A}\tilde{B} \\ \vdots & \vdots & \ddots & \vdots \\ \tilde{C}\tilde{A}^{N_p-1}\tilde{B} & \tilde{C}\tilde{A}^{N_p-1}\tilde{B} & \cdots & \tilde{C}\tilde{A}^{N_p-N_c-1}\tilde{B} \end{bmatrix}$$

Following the discretization of the kinematic equations of the point mass model in both horizontal and vertical directions:

$$v_x^{k+1} = v_x^k + a_{ex}^k T_s$$
$$v_y^{k+1} = v_y^k + a_{ey}^k T_s$$
$$x^{k+1} = x^k + v_x^k T_s \tag{8}$$
$$y^{k+1} = y^k + v_y^k T_s$$

Where $v_x$ and $v_y$ represent the velocities of the self-vehicle in the longitudinal and transverse directions, respectively. $x$ and $y$ denote the positions of the self-vehicle in the lateral and longitudinal directions along the simulated road. $a_{ex}$ and $a_{ey}$ denote the lateral and longitudinal accelerations of the self-vehicle, while $T_s$ represents the sampling period. The velocity and positions of the front and left vehicles are as follows:

$$\Delta v_{xi}^{k+1} = \Delta v_{xi}^k + (a_{xi}^k - a_{ex}^k)T_s$$
$$\Delta v_{yi}^{k+1} = \Delta v_{yi}^k + (a_{yi}^k - a_{ex}^k)T_s$$
$$\Delta x_i^{k+1} = \Delta x_i^k + \Delta v_{xi}^k T_s \tag{9}$$
$$\Delta y_i^{k+1} = \Delta y_i^k + \Delta v_{yi}^k T_s$$

Where, $\Delta v_{xi}$ and $\Delta v_{yi}$ denote the longitudinal and transverse velocity differences between the $i$ vehicle and the self-vehicle. $\Delta x_i$ and $\Delta y_i$ represent the longitudinal and transverse distances between the $i$ vehicle and the self-vehicle, respectively, while $a_{xi}$ and $a_{yi}$ signify the accelerations of the $i$ vehicle along the longitudinal and transverse axes.

## Obstacle avoidance trajectory planning

### Establishing vehicle driving conditions

Barrier-free overtaking trajectory planning necessitates the control of autonomous vehicles across varied driving conditions. Fig 3 illustrates typical real-world scenarios encountered by autonomous vehicles in everyday operations. In this scenario, Car1 is positioned ahead of the ego vehicle, while Car2 is in the adjacent left lane. The ego vehicle E modifies its actions—such as braking and overtaking—according to the behaviors of these vehicles and the surrounding conditions. The objectives of this trajectory planning include:

1. Ensuring the vehicle travels along the centerline of the lane.

2. Maintaining an appropriate travel speed.

3. Adhering to physical and dynamic constraints.

4. Avoid collisions with surrounding vehicles already at the lane boundary.

The architecture comprises two layers: the decision-making and trajectory-planning layers. The decision layer determines lane changes or braking and formulates these tasks as MPC constraints. A Finite State Machine (FSM) triggers specific activation functions based on constraints to execute various driving maneuvers. Throughout driving, real-time monitoring of nearby road occupancy and distances between the vehicle and the front and left vehicles ensures collision avoidance. Fig 1 illustrates that once the vehicle reaches a safe distance from

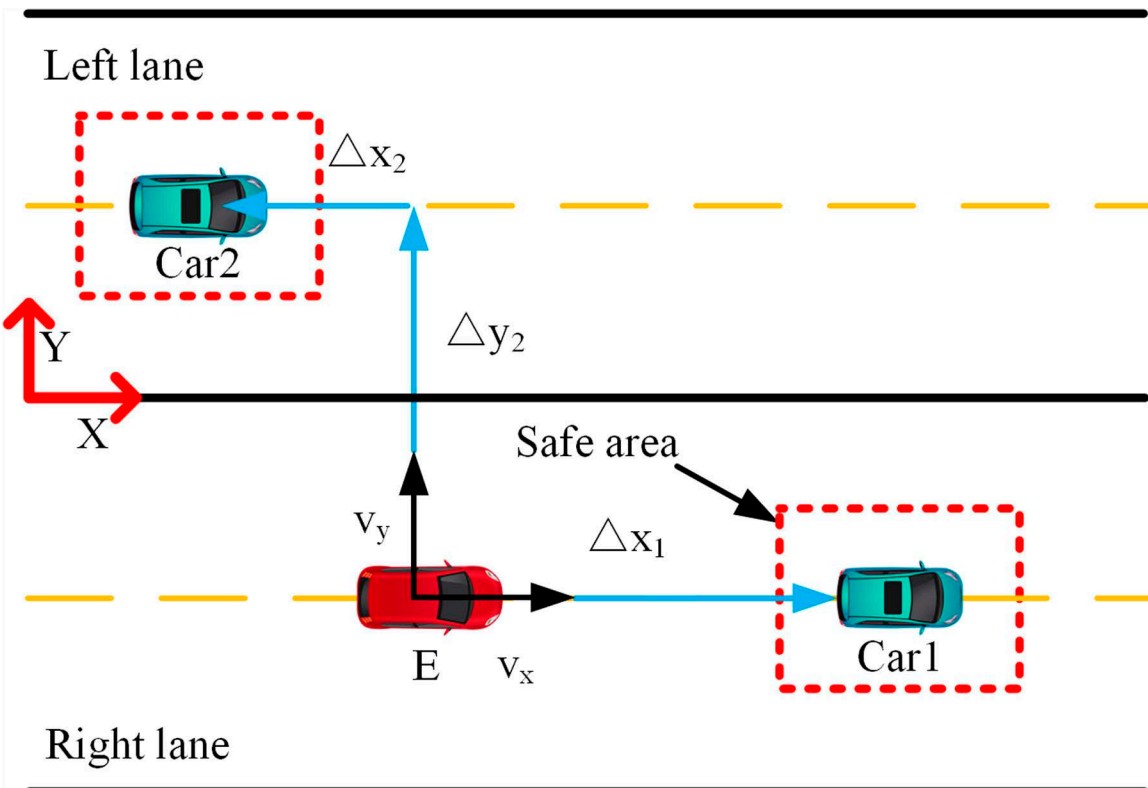

**Fig 3. Vehicle driving diagram.** The red vehicle represents the ego vehicle, the green vehicle denotes the obstacle vehicle, and the red border indicates the safety zone.

the preceding vehicle, it evaluates the availability of the left lane. If clear, the vehicle changes lanes; otherwise, it remains in its current lane. In case the front vehicle decelerates beyond a safe braking distance, the vehicle initiates braking until the left lane clears, then changes lanes, returning to its original lane after overtaking, contingent on conditions, thereby completing obstacle avoidance and overtaking maneuvers. The overtaking trajectory follows an S-shaped path that adjusts dynamically to real-time road conditions. A pre-optimized Sigmoid function adjusts constraints post-sampling for optimal trajectory planning. The MPC computes appropriate outputs $(a_{ex}, a_{ey})$ based on these constraints to execute optimal overtaking maneuvers. This study assumes knowledge of self-vehicle speed, positions of surrounding vehicles, and a straight road configuration.

## FSM-based decision-making

The FSM (Fig 4) algorithm determines braking or overtaking actions based on surrounding information, adjusting vehicle speed or lane position to ensure safety by considering adjacent lane usage, and the distance and speed of preceding vehicles. Depending on the scenario, it generates an appropriate activation function sent to the MPC to execute the corresponding driving response. The flowchart outlining this process is depicted below.

Where $\delta_k$ represents the lane change activation function; the vehicle initiates a lane change when $\delta_k = 1$. $\gamma_k$ denotes the braking activation function; braking occurs when $\gamma_k = 1$. This activation function guides subsequent Sigmoid constraint barriers, serves as input to the MPC for trajectory generation, and determines the transverse and longitudinal acceleration of the autonomous vehicle.

## Pre-optimization of the sigmoid function

The sigmoid function possesses the properties of boundedness, continuity, differentiability, and strict monotonicity, making it widely utilized in function-based trajectory planning algorithms. In this study, the Sigmoid function serves as an overtaking barrier in trajectory planning. The formulation of the Sigmoid function is given by [26–29]:

$$y(x) = \frac{y_{lat}}{1 + e^{-\zeta(x - \frac{s}{2})}} \tag{10}$$

Where $y_{lat}$ represents the lateral offset of the vehicle, $s/2$ denotes the longitudinal distance required to reach the symmetric path point $(s/2, y_{lat}/2)$ (this point is selected on the path to enhance smoothness and stability during path planning), and parameter $\zeta$ formally defines the function's curvature, where a higher value indicates sharper steering. To ensure the barrier function's safety and precision, preprocessing of the curvature parameter (Fig 5) can be integrated into the MPC.

The optimization of curvature form parameters is as follows. The formula for calculating curvature is:

$$\rho(x) = \frac{d\varphi(x)}{ds(x)} \tag{11}$$

Where:

$$ds(x) = \sqrt{(dx)^2 + (dy)^2}$$
$$\varphi(x) = \arctan(\frac{dy}{dx})$$

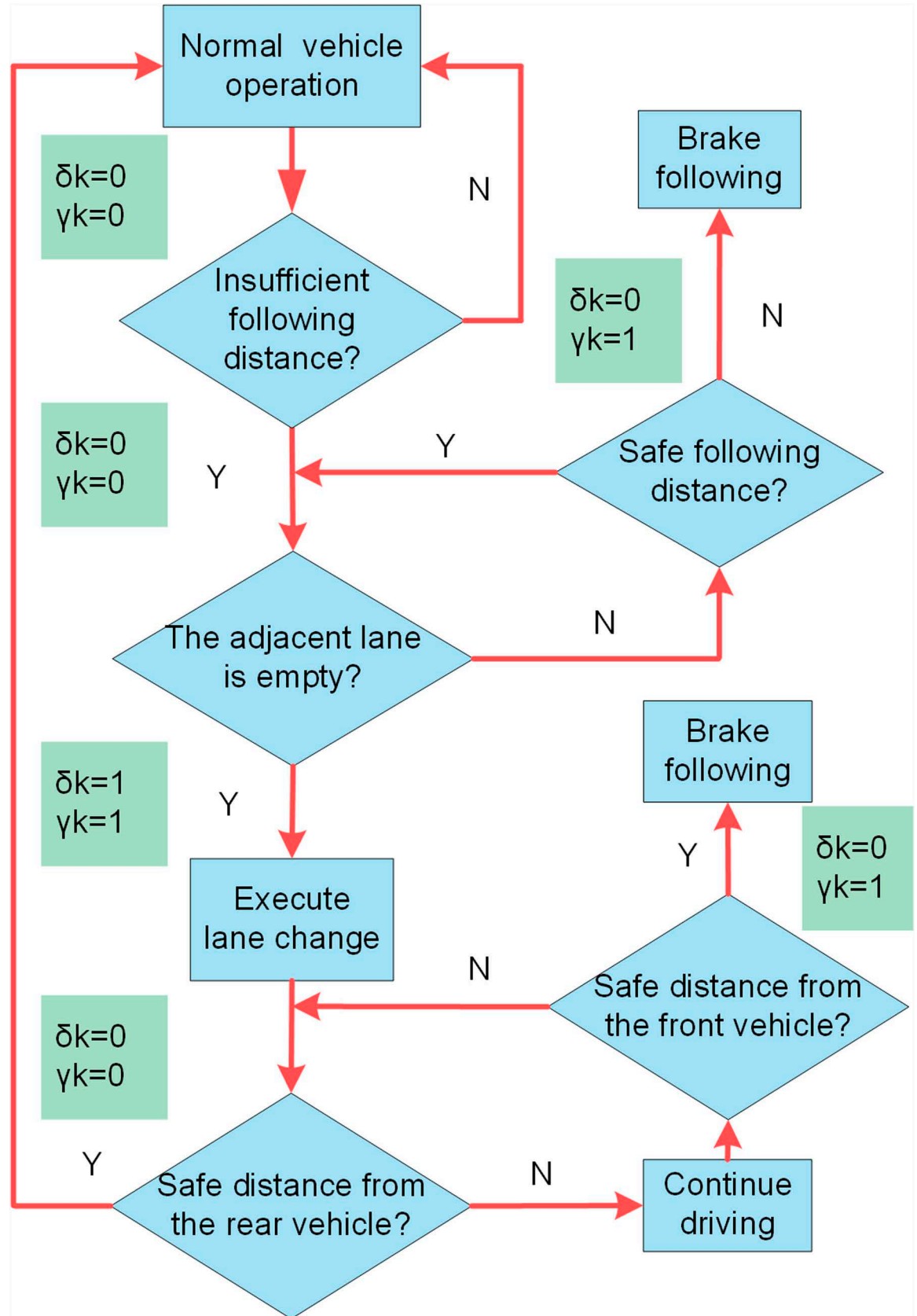

**Fig4. FSM flowchart.** Blue denotes the vehicle's operational processes, while green represents the decision parameters.

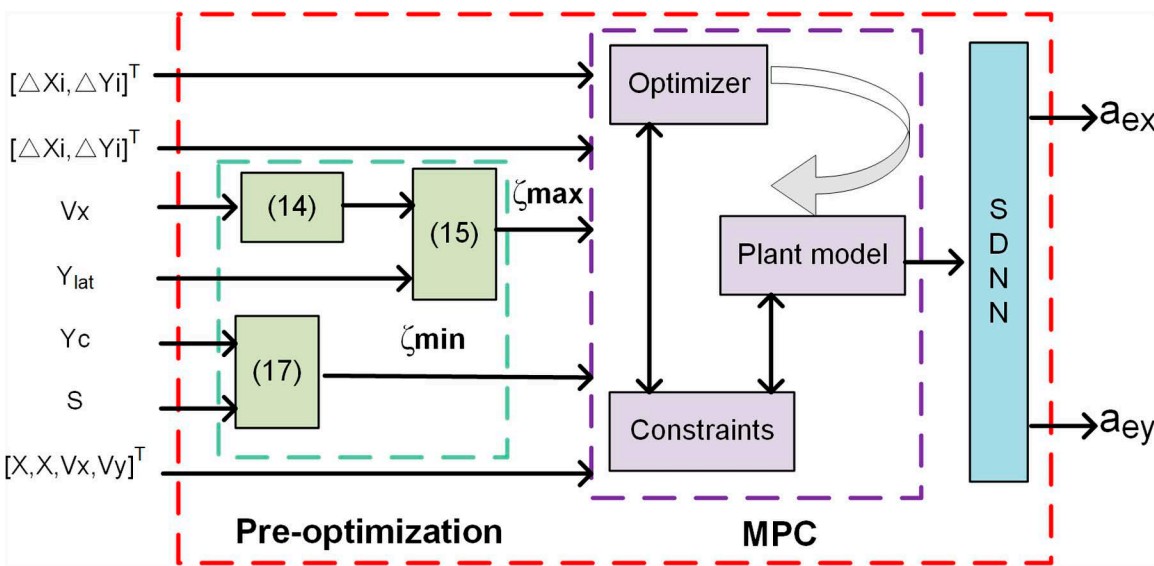

**Fig 5. Pre-optimization flowchart.** The left side represents inputs, while the right side represents outputs. Green denotes the pre-optimization of curvature parameters, and purple indicates the MPC operational process.

When $d(x)$ changes minimally, it can be approximated that $d(s) \approx d(x)$ and $\varphi(x) \approx \dfrac{d(y)}{d(x)}$. In conclusion, the formula for curvature calculation can be rewritten as:

$$\rho(x) = \frac{d\varphi(x)}{d(x)} \approx \frac{d^2 y(x)}{d(x)^2} \tag{12}$$

After calculation, the heading angle and curvature are [28]:

$$\varphi(x) = \frac{\zeta y_{lat} e^{-\zeta(x-\frac{s}{2})}}{(1 + e^{-\zeta(x-\frac{s}{2})})^2} \tag{13}$$

$$\rho(x) = \frac{\zeta^2 y_{lat}(e^{-\zeta(x-\frac{s}{2})} - 1)e^{-\zeta(x-\frac{s}{2})}}{(1 + e^{-\zeta(x-\frac{s}{2})})^4} \tag{14}$$

From the formula above, it is evident that the curvature depends on parameter $\zeta$, allowing modification of the Sigmoid constraint barrier's shape. Various lane-changing techniques exist with differing curvatures, yet adherence to physical and road safety constraints is paramount in vehicle travel. Thus, constraining lane change curvature within MPC involves integrating it as a set of constraints, where the curvature $\rho$ must adhere to limits defined between minimum $\rho_{min}$ and maximum $\rho_{max}$ curvatures.

The maximum curvature is obtained at the inflection point of the Sigmoid function, specifically at $\dfrac{d\rho(x)}{d(x)} = 0$. Substituting the determined inflection point $x$ value into $\rho(x)$ yields the maximum curvature value $\rho_{max}$. At this point, the curvature parameter $\zeta$ also achieves its maximum value:

$$\zeta_{max} = \sqrt{\frac{\rho_{max}(\gamma + 1)^3}{y_{lat}\gamma(\gamma - 1)}} \tag{15}$$

where $\gamma = e^{1.3170}$ is a constant determined through calculation $\zeta_{max}$. The maximum curvature of the safety trajectory must satisfy vehicle dynamics constraints. Steering should not exceed the maximum lateral acceleration, which is influenced by the friction between the vehicle and the road surface, as well as the front wheel steering angle. Lateral acceleration depends on the friction factor between the vehicle and the road surface, along with the front wheels' rotation angle. The upper limit of lateral acceleration is determined by $a_{y\max} = V_x \dot{\varphi}_{max}$, Where $\dot{\varphi}_{max} = 0.85\mu g / V_x$ determines the upper limit of yaw rate $\dot{\varphi}$. $V_x$ represents the longitudinal velocity of the vehicle, $\mu$ denotes the friction factor between the car and the ground, and $g$ stands for gravitational acceleration. For ensuring turn stability, the maximum curvature of the lane change trajectory correlates with the maximum yaw rate:

$$\rho_{max} = \frac{\dot{\varphi}_{max}}{V_x} \tag{16}$$

Unlike the maximum curvature $\rho_{max}$, the minimum curvature $\rho_{min}$ considers geometric constraints, specifically ensuring that collisions are avoided during vehicle overtaking, as detailed in reference [29].

$$\zeta_{min} = -2\frac{\log(\frac{y_c}{1 - y_c})}{s} \tag{17}$$

Where $y_c$ represents the permissible error to waypoint $(s, y_{lat})$. Optimization of $\zeta$ can be integrated into MPC, where $[\zeta_{min}, \zeta_{max}]$ is treated as constraints in each rolling optimization, thereby generating standardized overtaking trajectories.

## Cost function and constraints of trajectory planning

Objectives 1 and 2 involve driving along the lane centerline at an appropriate speed, formulated as a cost function:

$$J = \sum_{k=0}^{N_p - 1} \left\| X^k - X_{ref}^k \right\|_Q^2 + \sum_{k=0}^{N_c - 1} \left\| \Delta u^k \right\|_R^2 + \rho\varepsilon^2 \tag{18}$$

$$J_2 = \sum_{k=0}^{N_p - 1}\sum_{i=1}^{n_v} \left\| \zeta_i \right\|_\chi^2 \tag{19}$$

Where $X^k = [y^k, v_x^k]^T$ and $X_{ref}^k = [y_{ref}^k, v_{ref}^k]^T$. The input $\Delta u = [a_x^k, a_y^k]^T$ represents changes in the lateral and longitudinal acceleration of the vehicle. $J_2$ represents the cost function involving optimized slack variables, where $N_c$ denotes the control time domain, $N_p$ signifies the prediction time domain, $n_v = 1/2$ indicates the number of surrounding vehicles, and $\chi$ stands for the weight variable

To ensure both driving comfort and adherence to physical conditions, the hard constraints are:

$$\begin{cases} a_{ex}^k \in [a_{ex\min}, a_{ex\max}] \\ a_{ey}^k \in [a_{ey\min}, a_{ey\max}] \\ \Delta a_{ex}^k \in [\Delta a_{ex\min}, \Delta a_{ex\max}] \\ \Delta a_{ey}^k \in [\Delta a_{ey\min}, \Delta a_{ey\max}] \\ v_x \in [v_{x\min}, v_{x\max}] \end{cases} \tag{20}$$

The vehicle's longitudinal and lateral velocities are coupled to a certain extent. The slip angle $\beta$ constrains the relationship between longitudinal and lateral velocities [29], expressed as:

$$\beta = \arctan \frac{v_y}{v_x} \tag{21}$$

According to the small-angle approximation theorem, the above equation can be approximated as $\beta = \frac{v_y}{v_x}$. To ensure driving safety, the vehicle's slip angle must satisfy:

$$|\beta| < \beta_{max} \tag{22}$$

In summary, the longitudinal velocity $v_y$ can be constrained to be:

$$v_y^k \in [-v_x^k \beta_{max}, v_x^k \beta_{max}] \tag{23}$$

Road traveling constraints for:

$$y \in [y_{min} + \frac{L}{2}, y_{max} + \frac{L}{2}] \tag{24}$$

Where $L$ is the width of the car, $y_{max}$ and $y_{min}$ are the upper and lower boundaries of the road.

## Generation of sigmoid constraint barriers

In a realistic scenario of autonomous driving, each vehicle forms a Sigmoid constraint barrier. This S-type barrier function convexifies the navigable area around the ego vehicle, as illustrated in Fig 6.

For safety reasons, the autonomous vehicle must maintain a safe distance from the preceding vehicle and operate within a collision-free zone beyond the critical area. This depends on driving conditions and the decision-making module, which can be implemented through overtaking or braking maneuvers. When the ego vehicle approaches the hazardous zone of Car1, if the left lane is safe, a lane change may be executed. The feasibility of lane change is ensured by the S-shaped constraint barrier depicted in the figure. If the adjacent lane is occupied, the autonomous vehicle enters the danger zone (marked in red), making overtaking impossible (Fig 7). In such cases, it is necessary to adjust the speed and maintain a safe distance from the preceding vehicle.

The safety zone fluctuates over time and is contingent upon the state of the autonomous vehicle and the velocities of nearby vehicles. The Sigmoid constraint barrier allows the autonomous vehicle nearing a danger zone to maneuver into a collision-free zone. This mechanism is integrated into the MPC framework and triggered by an activation signal from the decision module [14].

$$\pm \Delta y_i^k \geq \frac{\delta^k y_{lat}}{1 + e^{-\zeta k(-\Delta x_i^k + S_f^k)}} \tag{25}$$

Where $y_{lat}$ is the lateral distance that should be retained between the self and the vehicle in front, which includes the width of the vehicle, $\zeta$ denotes the pre-optimized curvature geometry parameter; and $S_f$ stands for the longitudinal safety distance of the workshop, as defined by references [36,37]:

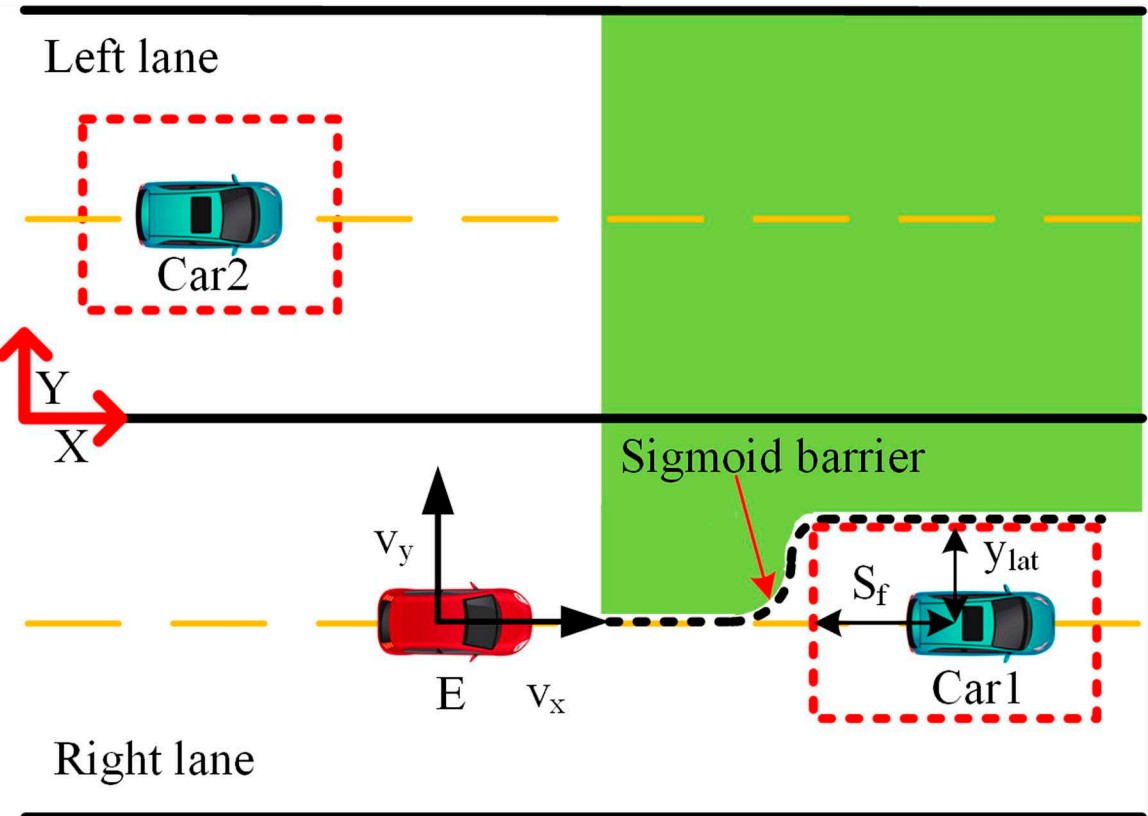

**Fig 6. Generate Sigmoid barrier.** The green region denotes the accessible area.

$$S_f = \frac{v_x^2}{2a_{e\max}} - \frac{v_{xi}^2}{2a_{i\max}} + v_x t + d_0 \tag{26}$$

Where $v_x$ represents the speed of the autonomous vehicle, $a_{e\max}$ denotes its acceleration, $v_{xi}$ signifies the speed of the obstacle vehicle, $a_{i\max}$ indicates the obstacle vehicle's acceleration, $t$ denotes the reaction time for peripheral information transmission, and $d_0$ denotes the minimum safe distance required. It should be noted that the sign of $\Delta y_i^k$ depends on the vehicle's lane position: it is positive when the autonomous vehicle is in the right lane and intends to change to the left lane, and negative when in the left lane changing to the right lane. Once in the overtaking lane, the vehicle can return to its original lane after ensuring a safe distance from the vehicle behind, completing the lane-change maneuver.

The FSM parameters are obtained from the following equation

$$\begin{cases} \left| \Delta x_1^k \right| \geq S_f - M\delta^k \\ \left| \Delta x_2^k \right| \geq S_f - M\gamma^k \end{cases} \tag{27}$$

## Simplified dual neural network for rapid MPC solution

In the rolling optimization process of MPC, the predicted output of the model can be expressed analytically and used to derive the optimal control sequence through quadratic programming [38]. Setting $E(k) = \phi\xi$ allows for the reformulation of Equation (18) as:

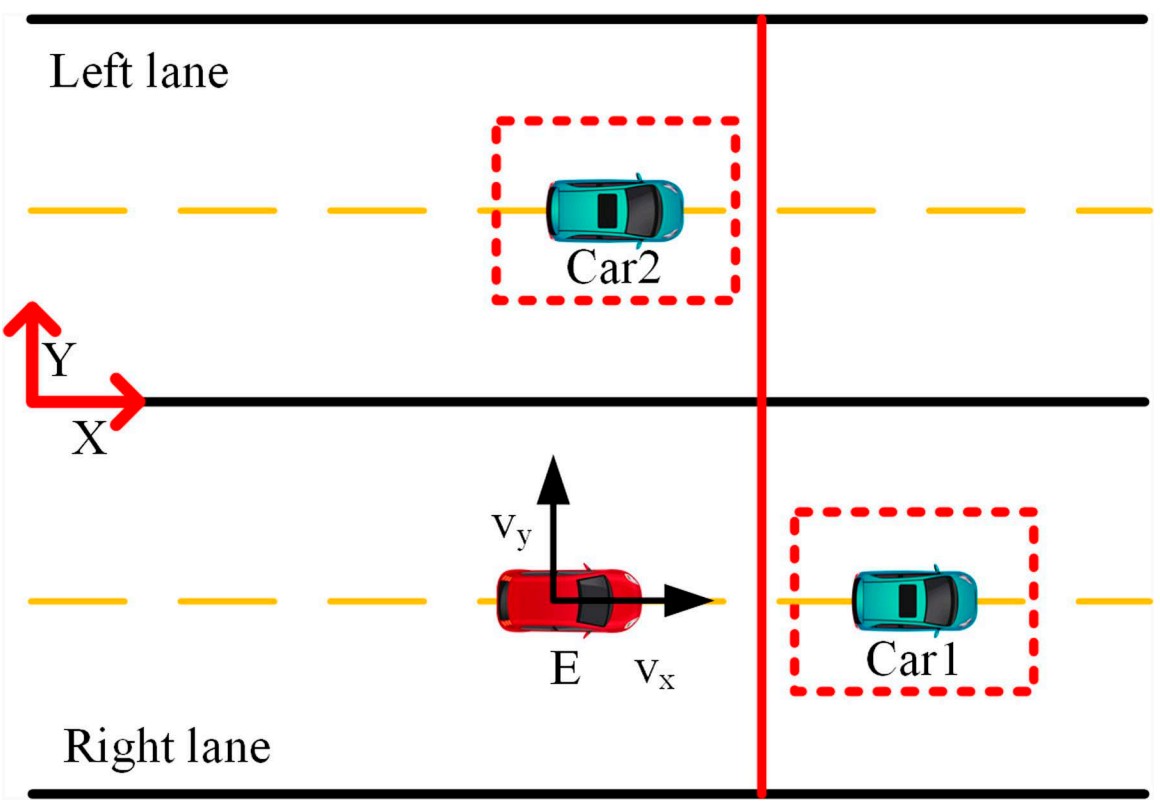

**Fig 7. Brake schematic diagram.** The ego vehicle must neither cross the safety red line nor change lanes.

$$J = \Delta u^T \left[ \theta^T Q \theta + R \right] \Delta u + 2 E^T Q \theta \Delta u + E^T R E + \varepsilon^T \rho \varepsilon \tag{28}$$

The optimal solution to the above equation yields:

$$\frac{\partial J_{QR,i}}{\partial_{\Delta u_i}} = 2 \left( \theta^T Q \theta + R \right) \Delta u + 2 E^T Q \theta \Delta u \tag{29}$$

Expressed in the form of a quadratic programming problem:

$$\min J = \frac{1}{2} \bar{U}^T H \bar{U} + f^T \bar{U}$$
$$s.t \ \ l \le K \bar{U} \le h \tag{30}$$

where $\bar{U} = [\Delta u \ \varepsilon]^T$, $H = \begin{bmatrix} 2 \left( \bar{B}^T Q B + R \right) \\ & 2\rho \end{bmatrix}$, $f^T = \begin{bmatrix} 2 E^T Q \theta & 0 \end{bmatrix}$, if $l_i = -\infty$ or $h_i = \infty$

corresponds to a one-sided inequality constraint; if $l_i = -\infty$ and $h_i = \infty$, the corresponding inequality constraint does not exist.

Based on the duality principle, the dual problem of the above equation can be formulated as:

$$\max_{\lambda_i, \lambda_h} -\frac{1}{2} \bar{U} H \bar{U} + l^T \lambda_i - h^T \lambda_h$$
$$s.t \ H \bar{U} + f - K^T \lambda_i + K^T \lambda_h = 0 \tag{31}$$

Where $\lambda_i, \lambda_h$ is the Lagrange multiplier, setting $v = \lambda_i - \lambda_h$ and substituting into the above equation yields:

$$H\bar{U} + f - K^T v = 0 \tag{32}$$

For the quadratic programming problem, its Karush-Kuhn-Tucker Conditions (KKT) conditions are:

$$H\bar{U} + c - E^T v = 0$$

$$\begin{cases} \left(K\bar{U}\right)_i = l_i & v_i > 0 \\ \left(K\bar{U}\right)_i = h_i & v_i < 0 \\ l_i \leq \left(K\bar{U}\right)_i \leq h_i & v_i = 0 \end{cases}$$

The above equation can be simplified to:

$$\begin{cases} H\bar{U} + f - K^T v = 0 \\ K\bar{U} = g\left(K\bar{U} - v\right) \end{cases} \tag{33}$$

Where $g(z) = \begin{bmatrix} g(z_1) & g(z_2) & \cdots & g(z_p) \end{bmatrix}^T$ is a piecewise linear vector function defined as follows:

$$g(zi) = \begin{cases} l_i & z_i < l_i \\ z_i & l_i \leq z_i \leq h_i \\ h_i & z_i > h_i \end{cases} \tag{34}$$

According to equation (33):

$$\bar{U} = Ov - p \tag{35}$$

Where $O = H^{-1}E^T, p = H^{-1}f$ .

Based on the above, the Simplified Dual Neural Network model (SSDN) can be constructed as follows:

State Equation:

$$\kappa \frac{d_v}{d_t} = -K\bar{U} + g(K\bar{U} - v) \tag{36}$$

Output equations:

$$\bar{U} = Ov - p \tag{37}$$

Where $v$ is the state variable; and $\kappa > 0$ is the scaling factor that regulates the speed of convergence of the simplified dual neural network model.

Let $z = K\bar{U} - v$ , and bring in the state equation can be obtained:

$$\kappa \frac{dz}{dt} = -KOz + (EO - I)g(z) - Kp \tag{38}$$

Where $z$ represents neurons, $g(z)$ denotes a nonlinear activation function and $KO - I$ is a symmetric weight coupling matrix. In this context, SSDN can be viewed as a periodic neural network model.

The above is the continuous SDNN model, and the discrete SDNN model by the Runge-Kutta method is as follows:

State Equation:

$$v^{(i+l)} = v^{(i)} + \frac{T_p}{6}\left(k_1 + 2k_2 + 2k_3 + k_4\right) \tag{39}$$

Output equations:

$$U_{QR} = Ov^i - p \tag{40}$$

Where $T_p$ denotes the step size and the rest of the notation is as follows:

$$\begin{cases} f(t,v) = \left[-K\bar{U} + g\left(K\bar{U} - v\right)\right]/\kappa \\ k_1 = f\left(t^{(i)}, v^{(i)}\right) \\ k_2 = f\left(t^{(i)} + \frac{T_p}{2}, v^{(i)} + \frac{T_p}{2}k_1\right) \\ k_3 = f\left(t^{(i)} + \frac{T_p}{2}, v^{(i)} + \frac{T_p}{2}k_2\right) \\ k_3 = f\left(t^{(i)} + T_p, v^{(i)} + T_p k_3\right) \end{cases}$$

This discretized SDNN can be applied in Model Predictive Control (MPC) for rolling optimization. Regarding the quadratic programming of $J$, the correspondence between the parameters of the discretized SDNN and those of the quadratic programming is as follows:

$$\begin{cases} \bar{U} = [\triangle u \ \varepsilon]^T \\ K = \begin{bmatrix} I_{2*2} & I \end{bmatrix}^T \\ f^T = \begin{bmatrix} 2E^T Q\theta & 0 \end{bmatrix} \\ H = \begin{bmatrix} 2\left(\bar{B}^T QB + R\right) & \\ & 2\rho \end{bmatrix} \\ l = \begin{bmatrix} u_{min}^T - u(k-1)^T & \triangle u_{min}^T \end{bmatrix} \\ h = \begin{bmatrix} u_{max}^T - u(k-1)^T & \triangle u_{max}^T \end{bmatrix} \end{cases} \tag{41}$$

Based on the preceding discussion, the primary metric for evaluating the solution efficiency of neural networks is computational complexity, as evidenced by equations (29) and (36). The number of neurons in the SDNN is directly related to the number of bilateral inequality constraints, with W and E being known parameters. Consequently, solving the MPC using SDNN does not require operations such as matrix decomposition or inversion. In comparison to traditional solution methods, SDNN exhibits lower computational complexity, thereby demanding fewer resources and delivering superior algorithmic efficiency.

The convergence of SDNN and its relationship with the optimal solution of quadratic programming can be theorized by [38]:

Theorem I: SDNN has stability in the sense of Lyapunov and converges globally to the equilibrium point $v^\star$.

Theorem II: The equilibrium output $\bar{U}^* = Ov^* - p$ of SDNN is the optimal solution of quadratic programming, where $v^*$ is the equilibrium state point of SDNN.

The comparison of solution speeds between SDNN and traditional methods under varying $N_p$ conditions is presented in Fig 8(a). As the number of iterations increases, the performance gap in solution speed between conventional methods and SDNN becomes more apparent. This is primarily attributed to the increased computational complexity of matrix inversion and decomposition in conventional methods. The use of SDNN leads to a substantial improvement in solution speed. As shown in Fig 8(b), compared to other variant solution methods, SDNN exhibits superior speed in solving the MPC, regardless of whether the system is single-input or multi-input, provided the step size remains fixed.

## Simulation analysis

This study employs a hardware-in-the-loop (HIL) setup based on the National Instruments LabVIEW RT platform to validate the effectiveness of the proposed IMPC further. The IPC2 is integrated with the environment simulation software SCANeR, which facilitates the driving environment. Veristand-RT invokes individual submodules, establishes connections between sub-signals, and deploys the model onto the lower computer. The real-time version of CarSim-RT, provided by CarSim, operates in real-time through frequency synchronization with the lower computer, closely approximating actual vehicle dynamics. CarSim-RT updates the vehicle's position, speed, and acceleration, and displays HIL simulation imagery. The lower computer encodes signals in CAN format for data transmission, and the Controller Area Network (CAN) bus facilitates data exchange between the various platforms. This HIL experiment sets the control cycle to 10 ms, with the CAN bus baud rate configured at 500 kbps. Vehicle parameters and driving information are provided in Tables 1 and 2.

This study examines four distinct scenarios of dual-lane driving to demonstrate the feasibility of the proposed method, which not only achieves overtaking objectives but also satisfies constraints on slip angle $\beta$ and yaw rate $\varphi$ to ensure lateral stability (Fig 9).

Simulation parameters are presented in the following Table 1.

The simulated vehicle driving environment information is presented in Table 2.

Driving information of the ego vehicle relative to surrounding vehicles in three driving scenarios.

1. There is a slower vehicle ahead, and a faster vehicle occupies the left lane.

The ego vehicle travels at 30 m/s. An obstacle vehicle in the left lane travels faster than the ego vehicle. The vehicle ahead moves slower than the ego vehicle, preventing safe overtaking. Initially, the ego vehicle applies brakes and waits for the left lane to clear before attempting to overtake. During overtaking, it first decelerates to 25 m/s without changing lanes. Once the left lane is clear, the ego vehicle changes lanes, ensuring a safe distance between vehicles ahead and behind. It then returns to its original lane and resumes traveling at 30 m/s, as illustrated in the ego vehicle trajectory diagram (Figs 10, 11, 12).

2. The preceding vehicle is obstructing, while the left lane remains vacant.

The vehicle travels at a speed of 30 m/s. The vehicle ahead decelerates, and the left lane is clear. When the vehicle approaches too closely to the preceding vehicle, it reduces its speed from 30 m/s to 28.89 m/s in preparation for overtaking. Upon entering the left lane, the vehicle maintains a safe following distance before merging back into the original lane and restoring its initial speed (Figs 13, 14, 15).

3. There is a slower vehicle ahead, and a slower vehicle occupies the left lane.

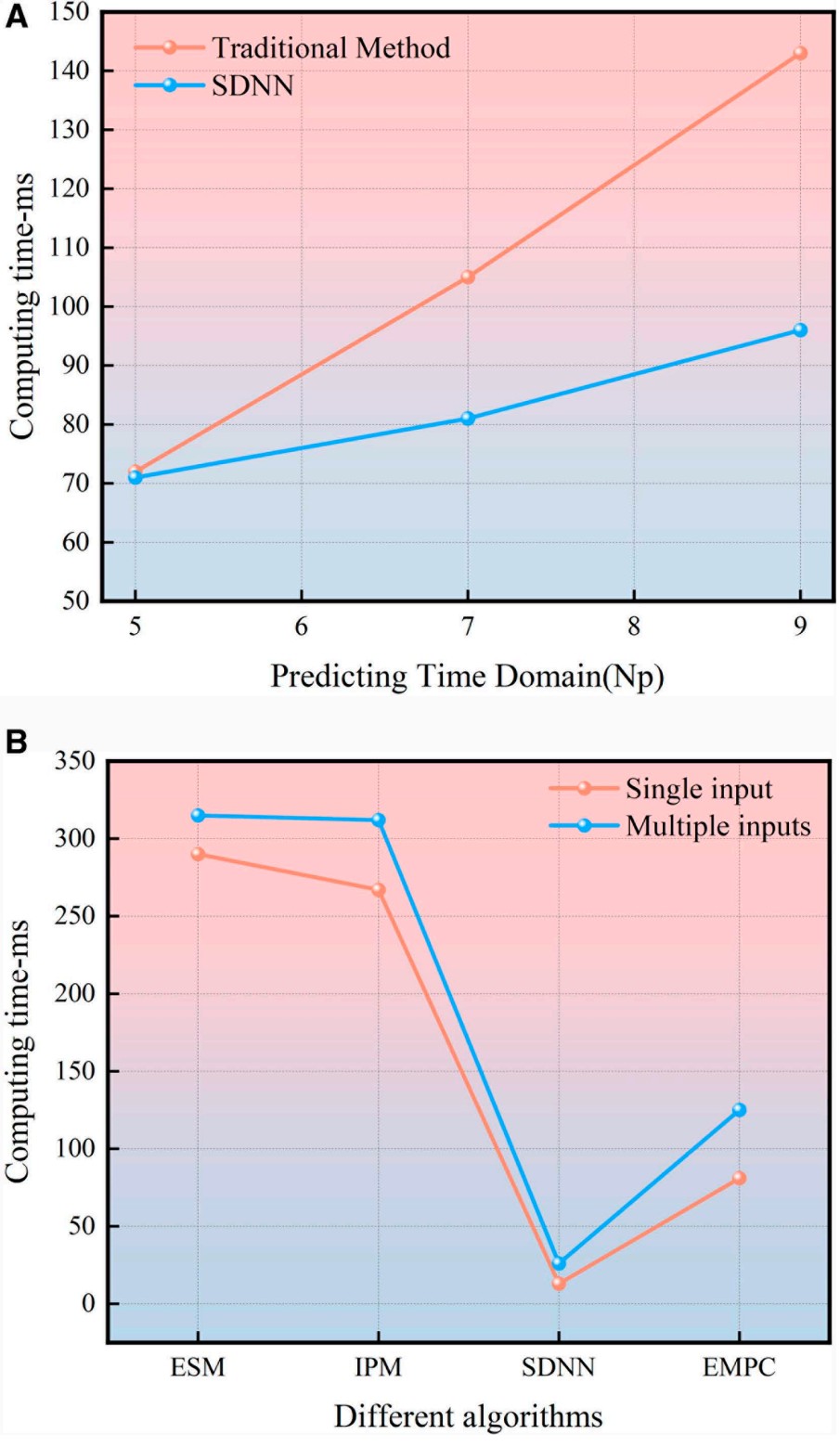

**Fig 8.** (a) **Comparison between SDNN and Traditional Methods.** The horizontal axis represents the prediction horizon, while the vertical axis denotes computation time; red denotes traditional methods, and blue represents SDNN. (b) **Comparison of different algorithms under different inputs.** The horizontal axis represents algorithm type and total surrounding computation speed, with red lines indicating single input and blue lines representing multiple inputs.

**Table 1. Simulation parameters.**

| Soft constraints limits | |
|---|---|
| $y_{\min} = 0\ m$ | $y_{\max} = 7.2\ m$ |
| $Vx_{\min} = 0\ m/s$ | $Vx_{\max} = 40\ m/s$ |
| $L = 2m$ | $\beta_{\max} = 10°$ |
| **Hard constraints limits** | |
| $a_{x\min} = -4\ m/s^2$ | $a_{x\max} = 2\ m/s^2$ |
| $a_{y\min} = -2\ m/s^2$ | $a_{y\max} = 2\ m/s^2$ |
| $\Delta a_{x\min} = -1.5\ m/s^2$ | $\Delta a_{x\max} = 1.5\ m/s^2$ |
| $\Delta a_{y\min} = -0.5\ m/s^2$ | $\Delta a_{y\max} = 0.5\ m/s^2$ |
| **Safety barrier** | |
| $y_{lat} = 3.6\ m$ | $d = 1.3\ m$ |

$L$ represents vehicle width, $y$ denotes road constraints, $y_{lat}$ indicates road width, and $d$ represents minimum inter-vehicle spacing.

**Table 2. Driving environment information.**

| Scenario | Vx (m/s) | X1 (m) | X2 (m) | Vx1 (m/s) | Vx2 (m/s) |
|---|---|---|---|---|---|
| 1 | 30 | 20 | -80 | 36 | 18 |
| 2 | 30 | 180 | – | 18 | – |
| 3 | 30 | 30 | -50 | 10 | 18 |

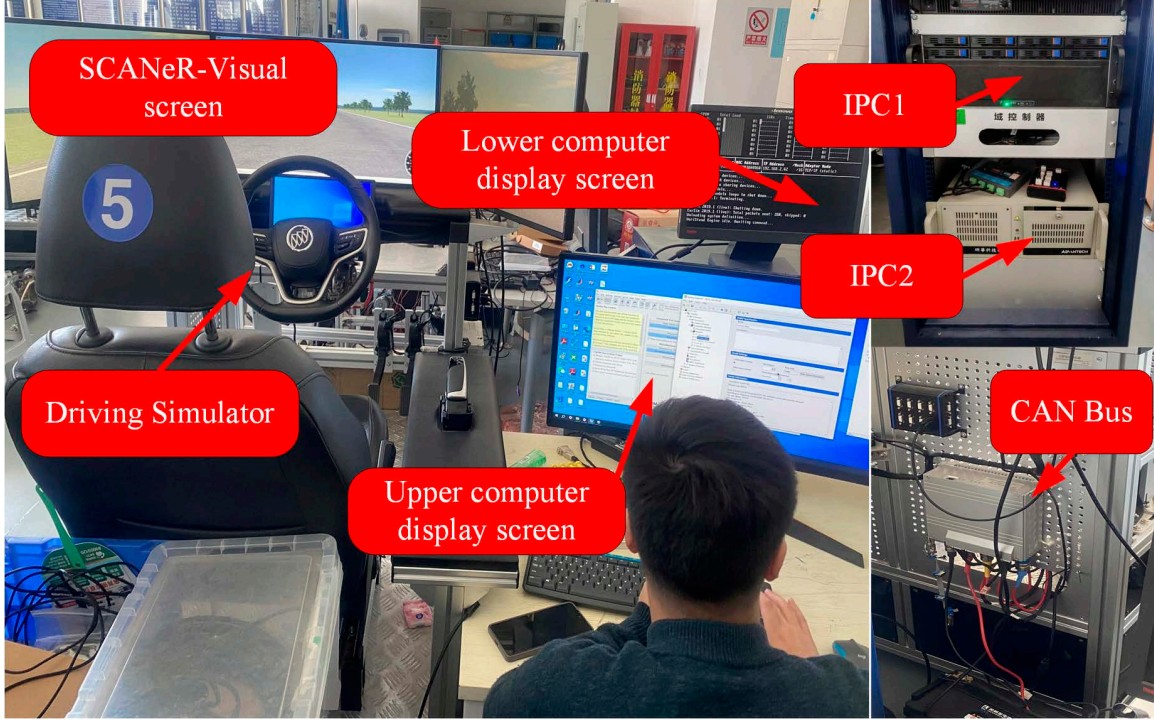

**Fig 9. Hardware in the loop experimental platform.** Showcasing our team's ongoing hardware in the loop experiment process.

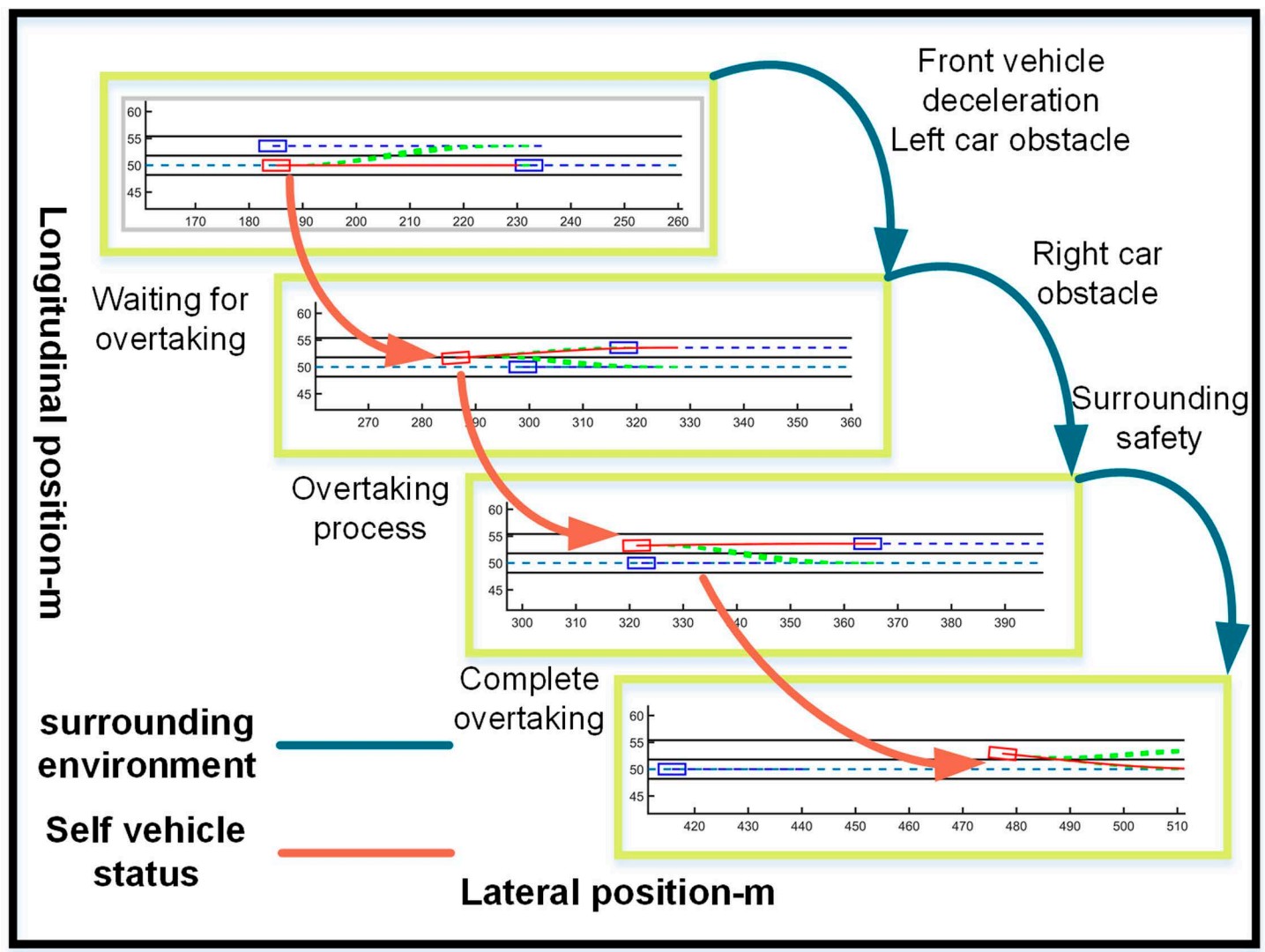

**Fig 10. Schematic diagram of the overtaking process.** Illustration depicting the changes in the ego vehicle's state relative to surrounding vehicles during the overtaking process.

The vehicle starts off traveling at 25 m/s. Encountering slower traffic ahead, it notices a faster but distant obstacle in the left lane when approaching too closely. The vehicle accelerates to 30 m/s to pass the slower traffic safely, then returns to its original lane, maintaining a speed of 30 m/s (Figs 16, 17, 18).

4. There is a slower vehicle ahead, and a faster vehicle occupies the left lane.(Wet and slippery road surface on rainy days)

The subject vehicle is traveling at a speed of 30 m/s. In the left lane, an obstacle vehicle is traveling at a speed greater than that of the subject vehicle. The vehicle ahead is traveling at a slower speed than the subject vehicle, preventing the subject vehicle from overtaking safely. The subject vehicle first applies the brakes and waits until the left lane is clear before attempting to overtake. The trajectory of the subject vehicle is depicted in the figure.

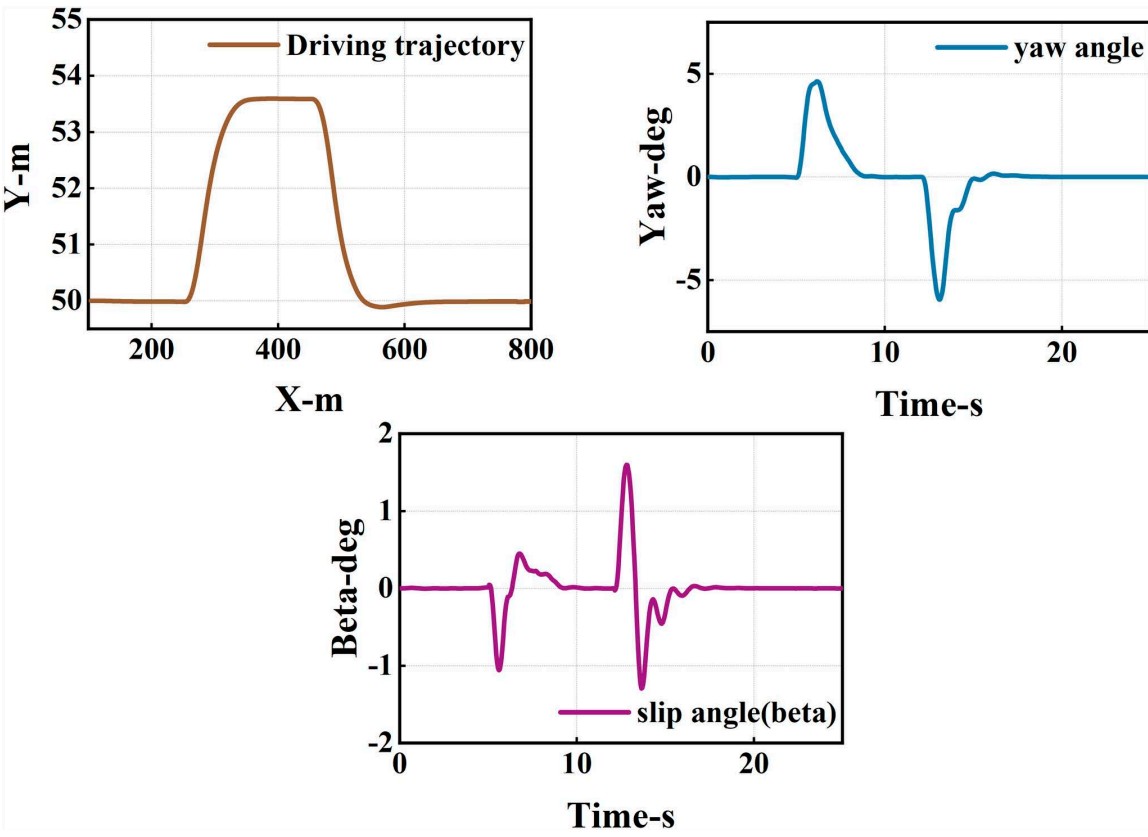

**Fig 11. Diagram of driving trajectory and steering angle.** The figures sequentially depict the trajectory, heading angle, and slip angle throughout the overtaking maneuver.

During the overtaking maneuver, the subject vehicle first reduces its speed to 25 m/s and refrains from changing lanes. Once the left lane is clear, the subject vehicle changes lanes. After confirming a safe distance between the subject vehicle and the vehicles in front and behind, the subject vehicle returns to its original lane and resumes a speed of 30 m/s. Compared to normal weather conditions, controlling the subject vehicle's speed during acceleration and deceleration is more challenging, and the speed curve becomes more pronounced (Figs 19–21).

## Conclusion

This paper proposes a collision-free trajectory planning method integrated with a decision-making process. The method utilizes an Improve Model Predictive Control framework; the Sigmoid function serves as a constraint barrier for vehicle dynamics. During the decision-making process, Finite State Machine parameters dynamically enable or disable constraints. Furthermore, the optimization of decision parameters and Sigmoid parameters is integrated into Model Predictive Control. Finally, Discrete Simplified Dual Neural Network is employed for rapid Quadratic Programming problem solution in MPC, obtaining optimal lateral and longitudinal accelerations, significantly simplifying the solving process. This approach enables timely driving decisions to prevent accidents.Additionally, the proposed method demonstrates adaptability and flexibility in various real-world driving scenarios.

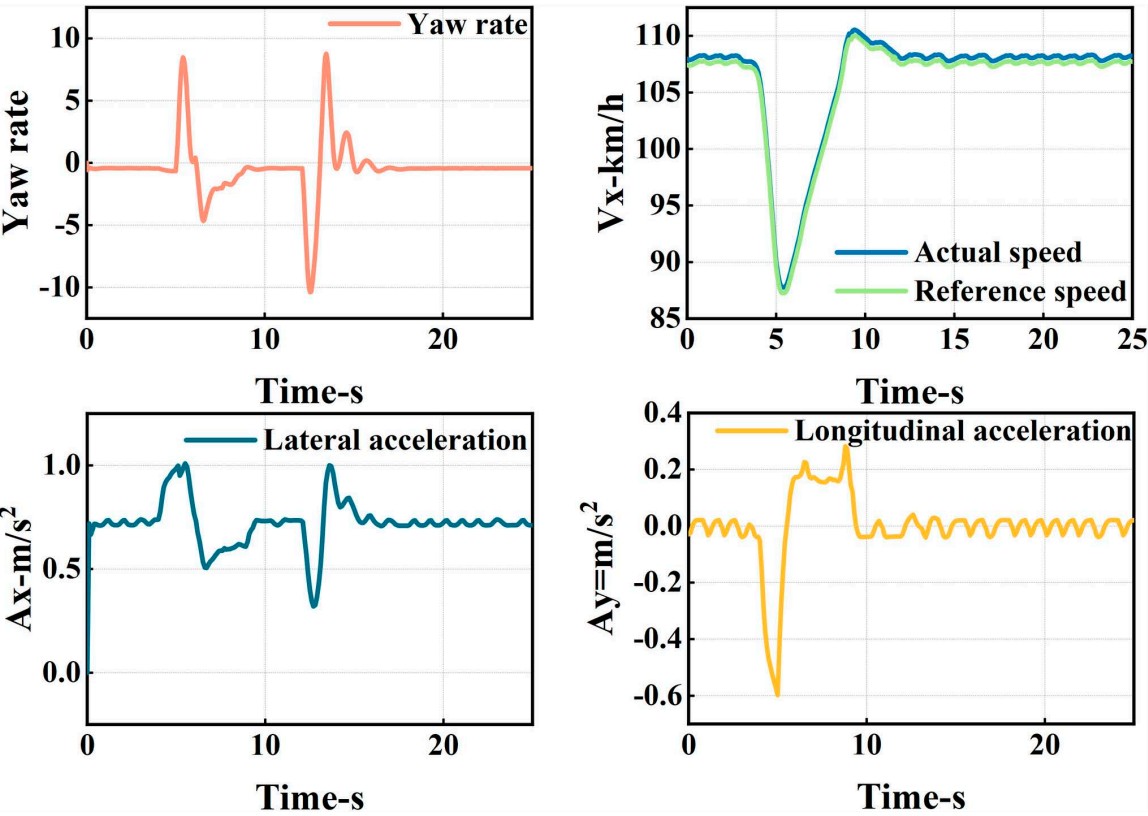

**Fig 12. Graph of acceleration, velocity, and angle change rate.** From top to bottom, the figures depict the images of yaw rate, velocity, lateral acceleration, and longitudinal acceleration during the overtaking process.

Simulation results validate method effectiveness in diverse driving environments, promising future real-time testing on actual vehicles.As the complexity of surrounding driving environments increases, challenges of obstacle avoidance and overtaking escalate. To address this, future work could explore priority-based algorithms in trajectory planning. Further investigations aim to enhance system performance and robustness.

## Acknowledgment

We would like to express our gratitude to Professor Haitao Ding for his laboratory simulation work. We express our gratitude to Professor Nianna Zhang for her support. We would like to extend our sincere appreciation to the editor and reviewers for their anticipated feedback and valuable suggestions on this manuscript.

## Author contributions

**Conceptualization:** Zonghao Li.

**Data curation:** Kai Sun.

**Formal analysis:** Heshu Zhang.

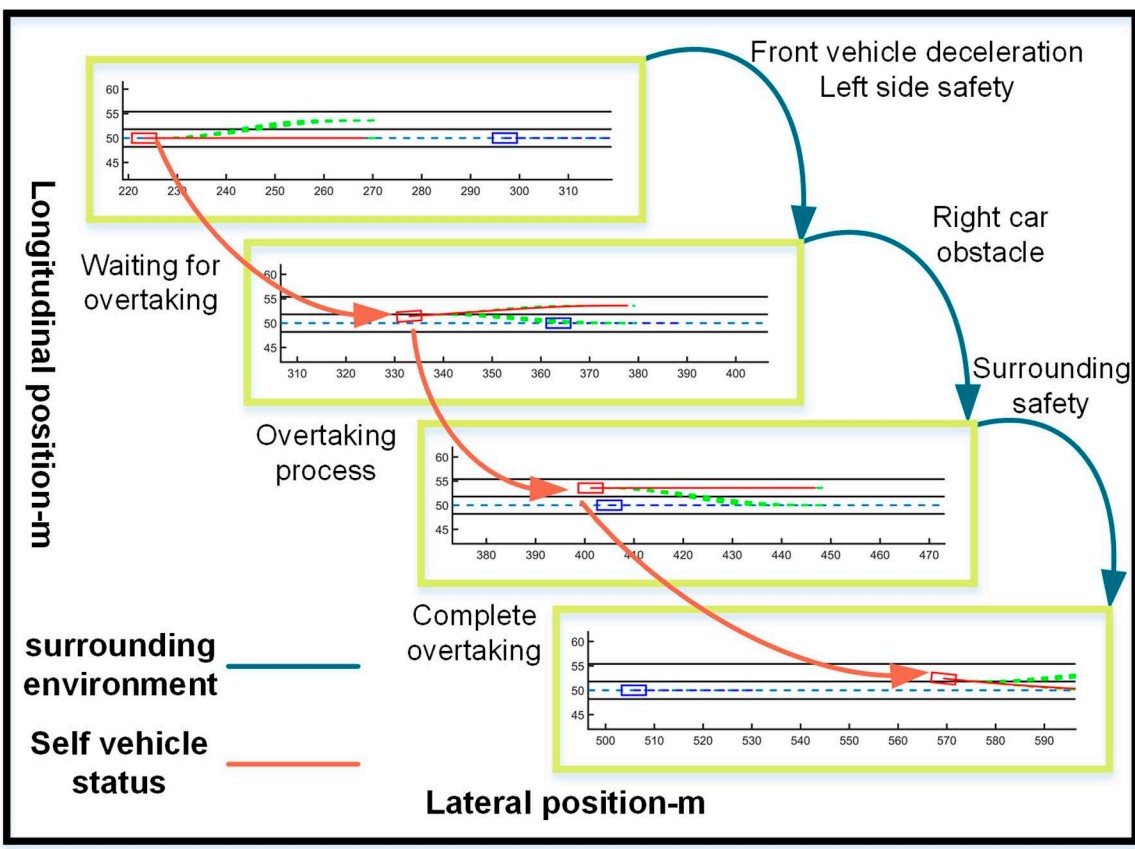

**Fig 13. Schematic diagram of the overtaking process.** Illustration depicting the changes in the ego vehicle's state relative to surrounding vehicles during the overtaking process.

**Funding acquisition:** Haitao Ding, Changhong Jiang.

**Investigation:** Kai Sun.

**Methodology:** Kai Sun.

**Project administration:** Niaona Zhang.

**Software:** Kai Sun.

**Validation:** Kai Sun.

**Visualization:** Kai Sun.

**Writing – original draft:** Kai Sun.

**Writing – review & editing:** Kai Sun.

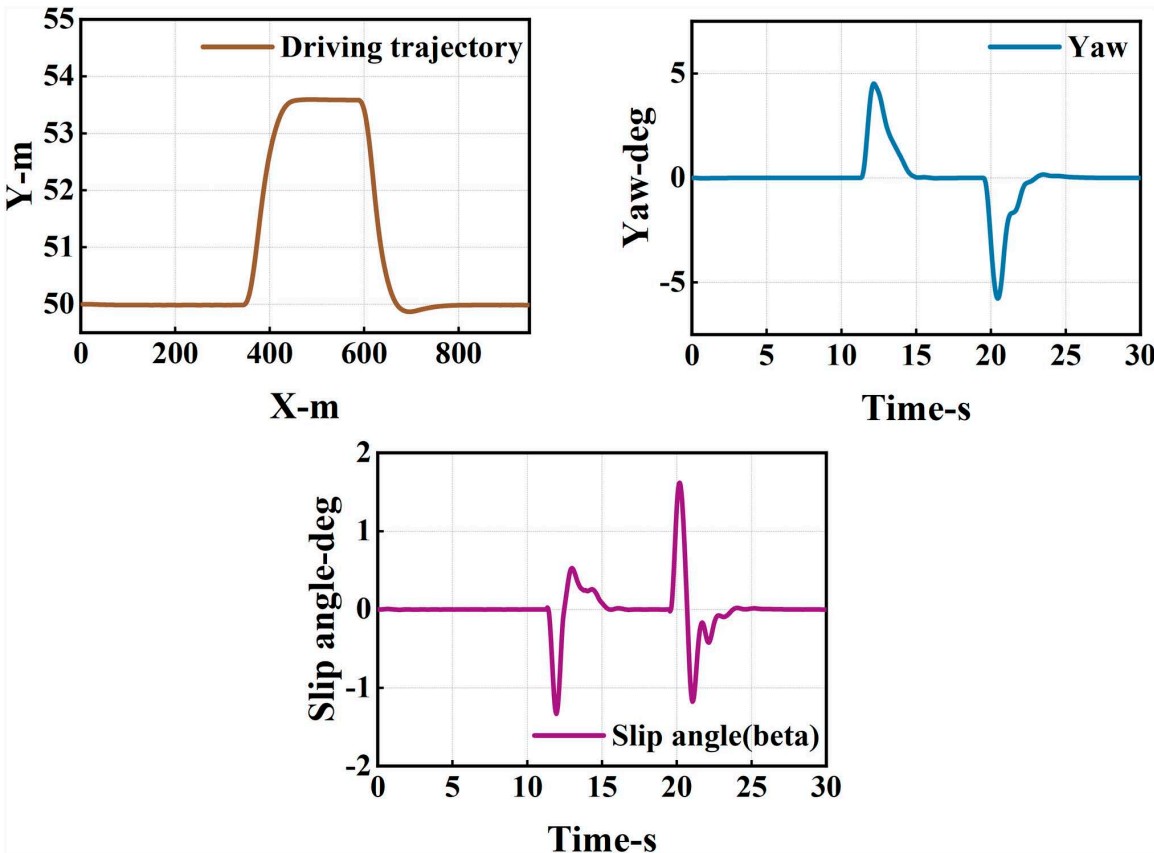

**Fig 14. Diagram of driving trajectory and steering angle.** The figures sequentially depict the trajectory, heading angle, and slip angle throughout the overtaking maneuver.

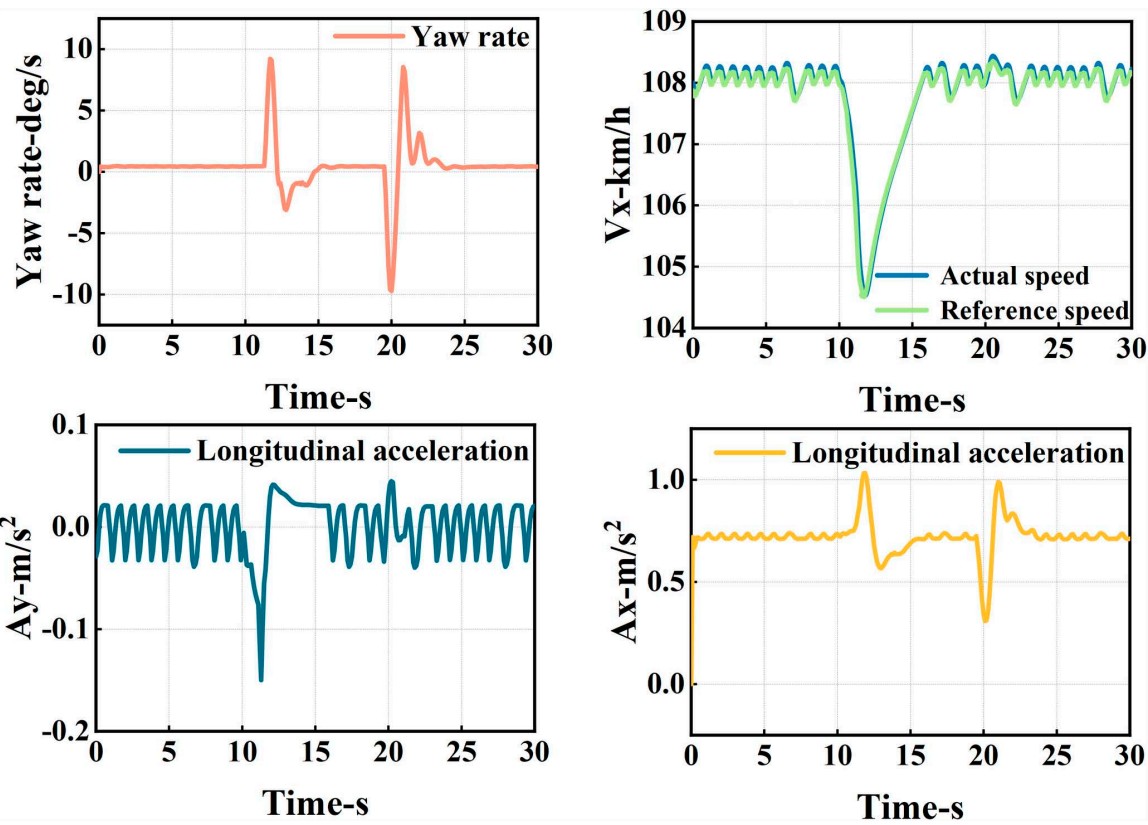

**Fig 15. Graph of acceleration, velocity, and angle change rate.** From top to bottom, the figures depict the images of yaw rate, velocity, lateral acceleration, and longitudinal acceleration during the overtaking process.

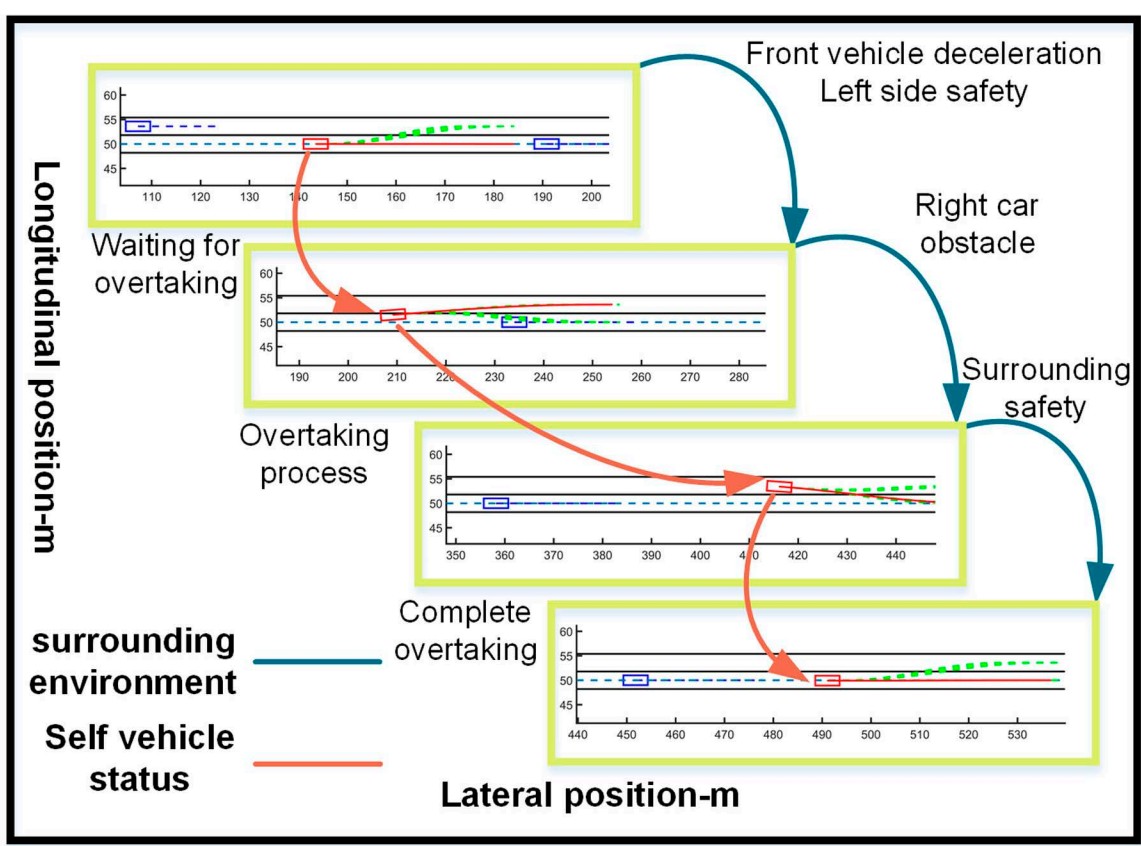

**Fig 16. Schematic diagram of the overtaking process.** Illustration depicting the changes in the ego vehicle's state relative to surrounding vehicles during the overtaking process.

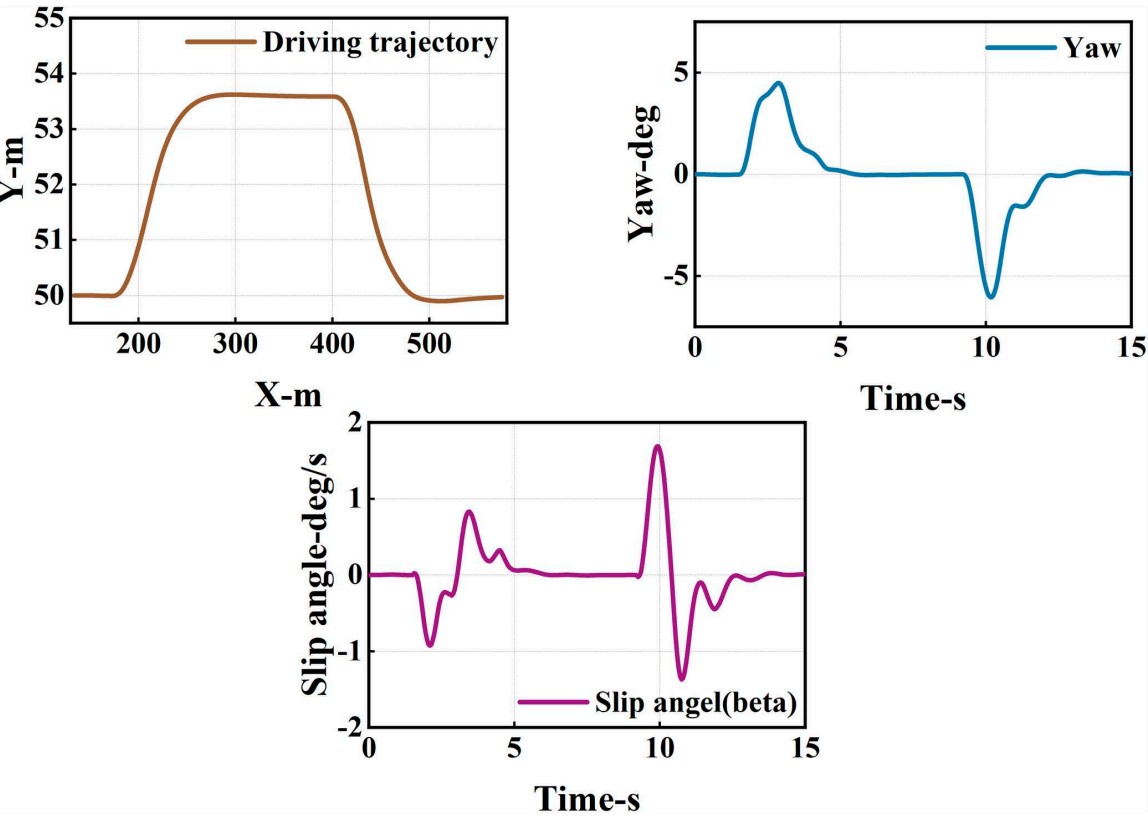

**Fig 17. Diagram of driving trajectory and steering angle.** The figures sequentially depict the trajectory, heading angle, and slip angle throughout the overtaking maneuver.

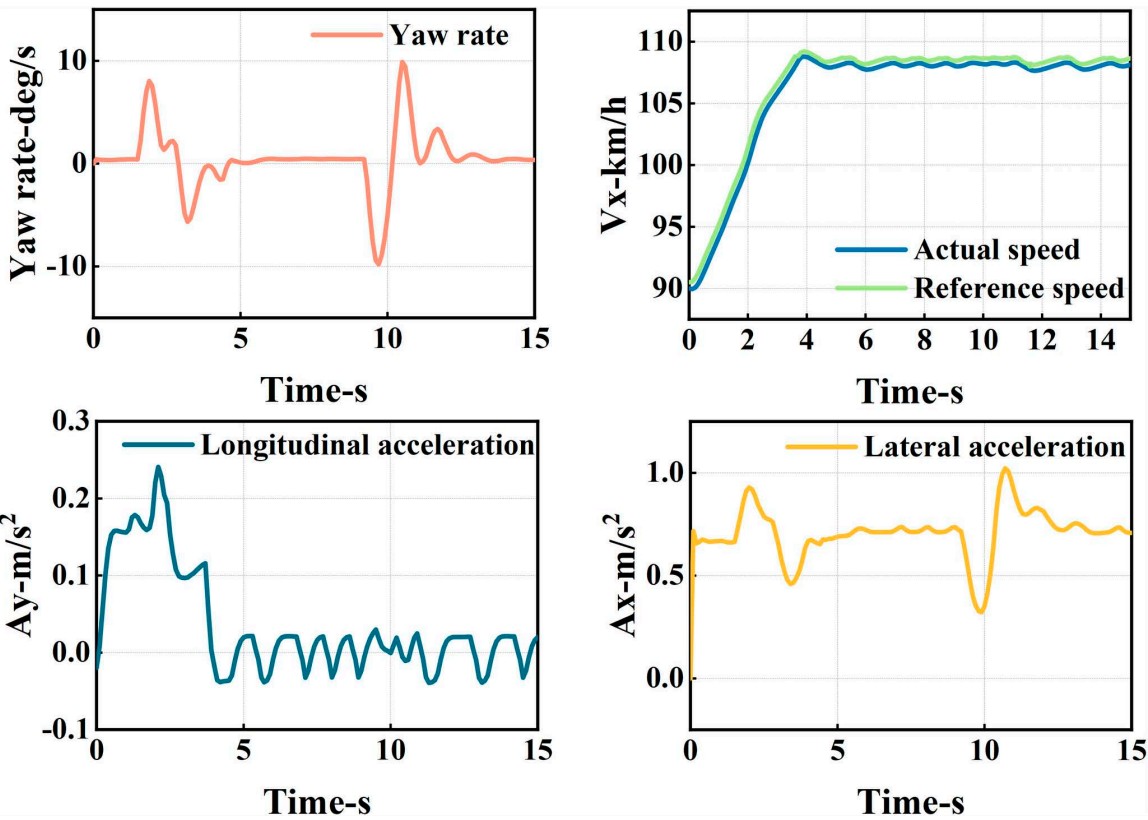

**Fig 18. Graph of acceleration, velocity, and angle change rate.** From top to bottom, the figures depict the images of yaw rate, velocity, lateral acceleration, and longitudinal acceleration during the overtaking process.

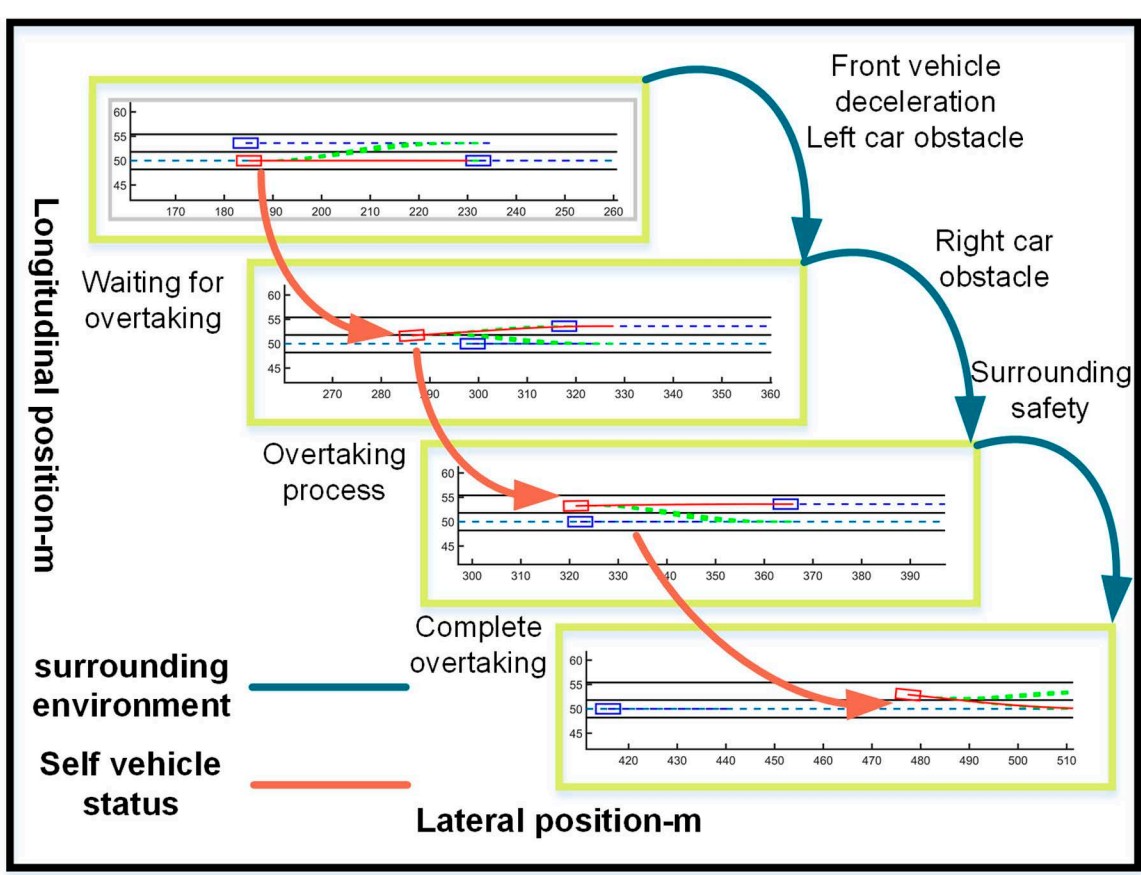

**Fig 19. Schematic diagram of the overtaking process.** Illustration depicting the changes in the ego vehicle's state relative to surrounding vehicles during the overtaking process.

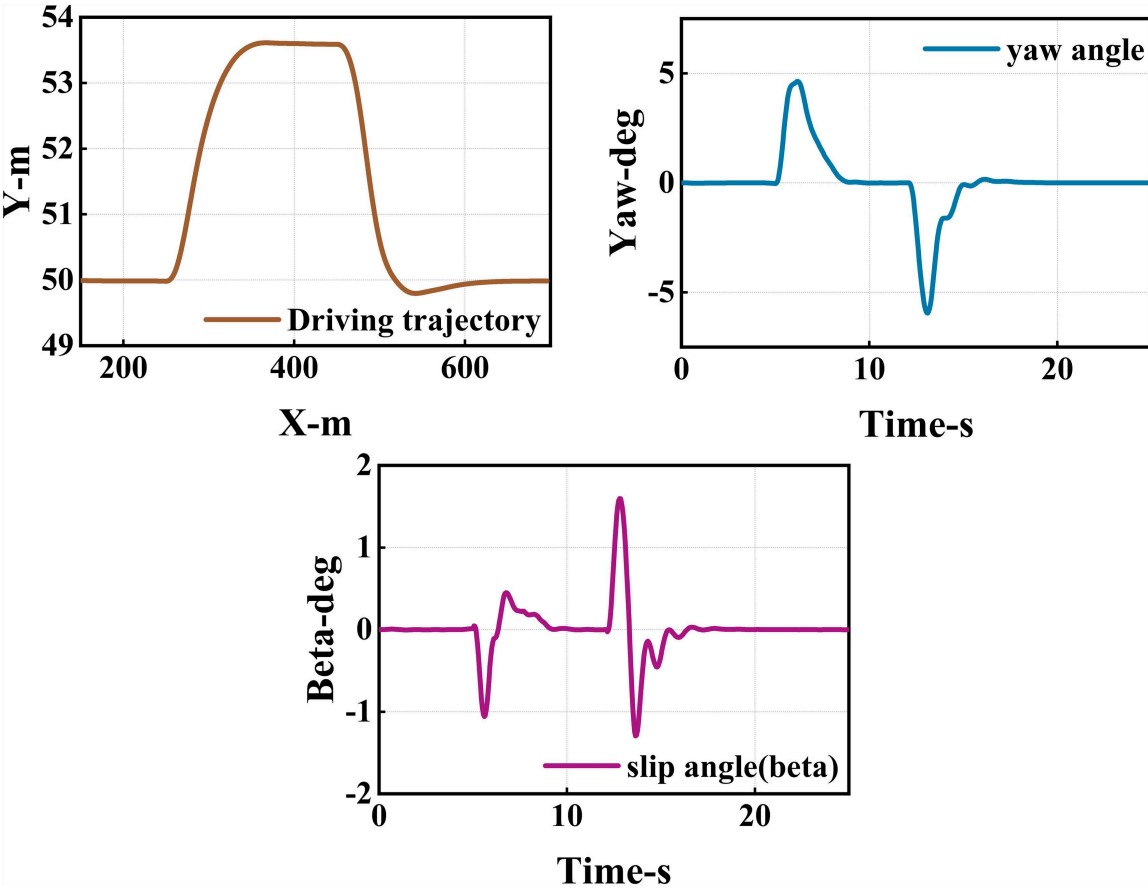

**Fig 20. Diagram of driving trajectory and steering angle.** The figures sequentially depict the trajectory, heading angle, and slip angle throughout the overtaking maneuver.

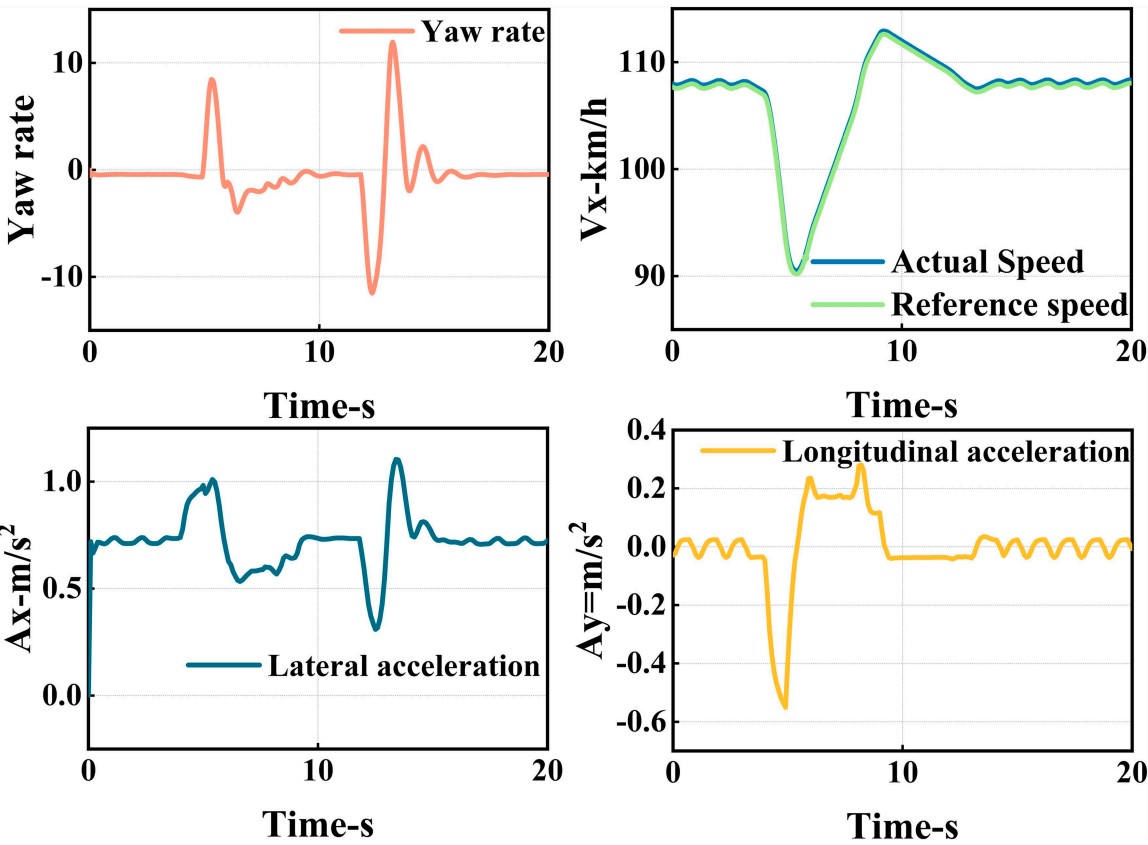

**Fig 21. Graph of acceleration, velocity, and angle change rate.** From top to bottom, the figures depict the images of yaw rate, velocity, lateral acceleration, and longitudinal acceleration during the overtaking process.

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
