## [Decision Letter · Decision Letter 0]

5 Dec 2024

PONE-D-24-30241Improved MPC for trajectory planning of self-driving carsPLOS ONE

Dear Dr. Zhang,

Thank you for submitting your manuscript to PLOS ONE. After careful consideration, we feel that it has merit but does not fully meet PLOS ONE’s publication criteria as it currently stands. Therefore, we invite you to submit a revised version of the manuscript that addresses the points raised during the review process.

We look forward to receiving your revised manuscript.

Kind regards,

Nishant Mukund Pawar, PhD

Academic Editor

PLOS ONE

**Journal Requirements:**

The authors would like to thank the financial support from the National Natural Science Foundation of China (Grant No. 51905045), the Science and Technology Development Plan Project of Jilin Province, China (Grant No. 20230508049RC), and the Open Fund of State Key Laboratory of Automotive Simulation and Control at Jilin University, China (Grant No. 20210237).

We would like to extend our sincere appreciation to the editor and reviewers for their anticipated feedback and valuable suggestions on this manuscript.

This research was funded by ZNN of the National Natural Science Joint Fund Project, grant number 51905045; ZNN of the Open Fund of the State Key Laboratory of Automotive Simulation and Control of Jilin University, grant number 20210237; and ZNN of the Jilin Province Science and Technology Development Plan Project, grant number 20230508049RC.

**Additional Editor Comments:**

Please see the review comments given by both the reviewers and address all the comments.

Reviewers' comments:

Reviewer's Responses to Questions

**Comments to the Author**

1. Is the manuscript technically sound, and do the data support the conclusions?

Reviewer #1: Yes

Reviewer #2: Yes

2. Has the statistical analysis been performed appropriately and rigorously? 

Reviewer #1: Yes

Reviewer #2: N/A

3. Have the authors made all data underlying the findings in their manuscript fully available?

Reviewer #1: Yes

Reviewer #2: Yes

4. Is the manuscript presented in an intelligible fashion and written in standard English?

Reviewer #1: Yes

Reviewer #2: Yes

5. Review Comments to the Author

**Reviewer #1:**  Review Comments PONE-D-24-30241

Summary of the Manuscript

The manuscript presents an Improved Model Predictive Control (IMPC) approach for trajectory planning in self-driving cars. The study integrates vehicle state and driving context to effectively execute maneuvers, incorporating a simplified model of vehicle dynamics, a Sigmoid function to restrict movements, and a Finite State Machine (FSM) for real-time decision-making. The approach also utilizes a discrete Simplified Dual Neural Network (SDNN) to solve Quadratic Programming (QP) problems rapidly. The IMPC is validated through integrated Carsim/Simulink simulations, demonstrating effective collision avoidance in various driving scenarios. Some good qualities of the manuscript are as follows:

1. Innovative Approach: The integration of SDNN for rapid QP problem-solving addresses the slow-solving issues of traditional MPC, making the approach more efficient.

2. Comprehensive Simulations: The use of Carsim/Simulink simulations provides a robust validation of the proposed method across various driving scenarios.

3. Clear Problem Definition: The manuscript clearly outlines the challenges associated with lane changes in diverse driving environments and the need for improved control methods.

4. Detailed Methodology: The step-by-step explanation of the IMPC approach, including the use of a Sigmoid function and FSM decision-making, is thorough and easy to follow.

5. Relevance: The focus on collision avoidance and overtaking maneuvers in autonomous vehicles is highly relevant given the current advancements in vehicle technology and AI.

Novelty and Contribution to the State of the Art

• Novelty: The use of SDNN to solve QP problems rapidly and the integration of a Sigmoid function for trajectory constraints are novel contributions that enhance the efficiency and robustness of MPC in trajectory planning for self-driving cars.

• Contribution: The manuscript significantly contributes to the state of the art by addressing the computational efficiency issues of traditional MPC and providing a validated approach for real-time trajectory planning and collision avoidance in autonomous vehicles.

Limitations of the Work

1. Limited Real-World Testing: The study heavily relies on simulations, and there is a lack of real-world testing to validate the practical applicability of the IMPC approach. The validation through Carsim/Simulink simulations, while comprehensive, does not address potential issues that may arise in real-world implementations, such as sensor noise and unexpected obstacles. How does the proposed IMPC approach perform in real-world driving conditions compared to controlled simulations?

2. Assumptions on Driving Scenarios: The scenarios considered might not cover all possible real-world driving conditions, which could limit the generalizability of the results. The simplified model assumes ideal driving conditions which may not reflect the complexity of real-world scenarios, potentially affecting the accuracy of the proposed approach. Can the IMPC approach handle more complex driving scenarios involving multiple dynamic obstacles and varying road conditions?

3. Computational Complexity: Although the manuscript claims to address computational efficiency, the actual computational load of the proposed method in real-time applications is not thoroughly discussed. What are the computational requirements for implementing the IMPC in real-time on an autonomous vehicle platform?

Points to Enhance the Quality of Work

Some possible points which can be included to enhance the quality of work are as follows. If not within the scope of present work the authors should include these within the future scope of work:

1. Incorporate Real-World Testing: The authors may conduct experiments in real-world driving environments to validate the effectiveness and robustness of the IMPC approach.

2. Expand Scenario Coverage: The authors can include a wider range of driving scenarios in simulations to test the generalizability of the approach.

3. Detail Computational Efficiency: The authors may provide a more in-depth analysis of the computational requirements and efficiency of the IMPC approach in real-time applications.

Grammatical Errors

• Sentence Structure: Some sentences in the manuscript are lengthy and could benefit from restructuring for clarity. For example, the sentence "The selection of MPC [14] is justified for several reasons: MPC can integrate road boundary constraints, vehicle dynamic constraints, and passable area constraints into its framework" can be simplified for better readability.

• Punctuation: There are instances of missing commas which can affect the readability. For example, "Moreover a discrete Simplified Dual Neural Network (SDNN) is employed" should be "Moreover, a discrete Simplified Dual Neural Network (SDNN) is employed."

**Reviewer #2: ** The study presents a comprehensive approach to enhancing the safety and efficiency of autonomous driving systems through advanced trajectory planning and control strategies. The integration of Model Predictive Control (MPC) with a Finite State Machine (FSM) for managing lane changes and braking maneuvers represents a significant advancement, offering a systematic method for real-time obstacle avoidance and maneuver execution. However, I would like to highlight a few concerns that could improve the manuscript's clarity and understanding.

1. Point mass model:

• The point mass model employed in this study simplifies vehicle motion by disregarding dimensions and dynamics, such as load transfer during acceleration, braking, and turning. I understand the authors used this to reduce computational complexity, however I’m wondering about its applicability in mixed traffic conditions, especially where vehicles of varying sizes and dynamics interact closely. Such interactions are common in diverse traffic environments, and ignoring these factors may lead to inaccuracies in predicting safe distances or potential collisions. Could you clarify how the point mass model accounts for or mitigates these potential inaccuracies, particularly in scenarios involving mixed and non-lane-based traffic, where diverse vehicle interactions are prevalent? Further, how was the model validated to ensure reliability in such complex traffic conditions?

• The omission of longitudinal and lateral load transfer effects may reduce the model’s accuracy in scenarios involving surrounding vehicles’ sharp turns, sudden braking, or acceleration, where these dynamics significantly influence vehicle performance. How does the model address scenarios require precise handling of dynamic effects, such as load transfer? Were any simplifications validated against real-world data to ensure their suitability for complex maneuvers?

• As real-world traffic scenarios often include challenges such as poor lighting, adverse weather conditions (while reading speed limits due to poor lighting), and unexpected obstacles (might not be of this study’s context), such as animals or pedestrians suddenly crossing the road. These factors can significantly impact the accuracy and reliability of motion predictions. In such a situation how does the model perform? Similarly, how does the model handle scenarios involving sudden, unpredictable events as mentioned earlier.

2. The study utilized a decision-making framework for lane changes and braking based on predefined constraints and a finite state machine (FSM). Although this approach can be efficient, it may not be effective in addressing complex or rapidly changing traffic conditions, such as heavy congestion or unpredictable driver behavior. Is there any better adaptive decision-making mechanism (e.g., reinforcement learning or data-driven methods) to dynamically respond to varying traffic scenarios. Incorporating predictive modeling of surrounding vehicle behaviors could significantly improve safety and effectiveness.

6. PLOS authors have the option to publish the peer review history of their article (what does this mean? ). If published, this will include your full peer review and any attached files.

**Do you want your identity to be public for this peer review?** For information about this choice, including consent withdrawal, please see our Privacy Policy .

Reviewer #1: No

Reviewer #2: No

---

## [Author Response · Author response to Decision Letter 0]

24 Dec 2024

Dear Reviewer,

Thank you for your careful review of our manuscript. Regarding the questions you raised, we have addressed them one by one in the "Response to Reviewers" section. We hope our responses will satisfy you.

Best regards,

Kai Sun

---

## [Decision Letter · Decision Letter 1]

18 Feb 2025

Improved MPC for trajectory planning of self-driving cars

PONE-D-24-30241R1

Dear Dr. Niaona Zhang,

We’re pleased to inform you that your manuscript has been judged scientifically suitable for publication and will be formally accepted for publication once it meets all outstanding technical requirements.

Kind regards,

Nishant Mukund Pawar, PhD

Academic Editor

PLOS ONE

Additional Editor Comments (optional):

Reviewers' comments:

Reviewer's Responses to Questions

**Comments to the Author**

1. If the authors have adequately addressed your comments raised in a previous round of review and you feel that this manuscript is now acceptable for publication, you may indicate that here to bypass the “Comments to the Author” section, enter your conflict of interest statement in the “Confidential to Editor” section, and submit your "Accept" recommendation.

Reviewer #1: All comments have been addressed

2. Is the manuscript technically sound, and do the data support the conclusions?

Reviewer #1: Yes

3. Has the statistical analysis been performed appropriately and rigorously? 

Reviewer #1: Yes

4. Have the authors made all data underlying the findings in their manuscript fully available?

Reviewer #1: Yes

5. Is the manuscript presented in an intelligible fashion and written in standard English?

Reviewer #1: Yes

6. Review Comments to the Author

Reviewer #1: The authors have appropriately addressed the comments and made necessary changes to the manuscript. The submitted version seems adequate and may be accepted for publication.

7. PLOS authors have the option to publish the peer review history of their article (what does this mean? ). If published, this will include your full peer review and any attached files.

**Do you want your identity to be public for this peer review?** For information about this choice, including consent withdrawal, please see our Privacy Policy .

Reviewer #1: No

---

## [Editor Report · Acceptance letter]

PONE-D-24-30241R1

PLOS ONE

Dear Dr. Zhang,

I'm pleased to inform you that your manuscript has been deemed suitable for publication in PLOS ONE. Congratulations! Your manuscript is now being handed over to our production team.

Kind regards,

on behalf of

Dr. Nishant Mukund Pawar

Academic Editor

PLOS ONE